# When does a Fermi puddle become a Fermi sea? Emergence of Pairing in Two-Dimensional Trapped Mesoscopic Fermi Gases

Emma **K.** Laird[1,*], Brendan **C.** Mulkerin[2], Jia Wang[3], and Matthew **J.** Davis[1]

**1** ARC Centre of Excellence in Future Low-Energy Electronics and Technologies, University of Queensland, Saint Lucia, Queensland 4072, Australia
**2** ARC Centre of Excellence in Future Low-Energy Electronics and Technologies, Monash University, Clayton, Victoria 3800, Australia
**3** Centre for Quantum Technology and Theory, Swinburne University of Technology, Hawthorn, Victoria 3122, Australia

* e.laird@uq.edu.au

## Abstract

**Pairing lies at the heart of superfluidity in fermionic systems. Motivated by recent experiments in mesoscopic Fermi gases, we study up to six fermionic atoms with equal masses and equal populations in two different spin states, confined in a quasi-two-dimensional harmonic trap. We couple a stochastic variational approach with the use of an explicitly correlated Gaussian basis set, which enables us to obtain highly accurate energies and structural properties. Utilising two-dimensional two-body scattering theory with a finite-range Gaussian interaction potential, we tune the effective range to model realistic quasi-two-dimensional scattering. We calculate the excitation spectrum, pair correlation function, and number of pairs as a function of increasing attractive interaction strength. For up to six fermions in the ground state, we find that opposite spin and momentum pairing is maximised well below the Fermi surface in momentum space. By contrast, corresponding experiments on twelve fermions have found that pairing is maximal at the Fermi surface and strongly suppressed beneath [M. Holten et al., Nature 606, 287–291 (2022)]. This suggests that the Fermi sea — which acts to suppress pairing at low momenta via Pauli blocking — emerges in the transition from six to twelve particles.**

# 1  Introduction

Fermionic superfluidity is a many-body phenomenon occurring in systems as diverse as liquid helium-three, superconductors, nuclear matter, neutron stars, and ultracold quantum gases. The key commonalities in these systems — that they flow without dissipation, have a non-classical rotational moment of inertia, and feature an energy gap in their elementary excitation spectrum — arise due to the pairing of fermions. Quantum gases provide an ideal experimental arena in which to interrogate the nature of fermion pairing since many of their degrees of freedom are highly tunable. Factors such as the number of particles, their internal states and interactions, the system dimensionality, and the confinement geometry can all be precisely measured and controlled [1–3]. In ultracold atomic Fermi gases, this has led to the realisation and detailed study of the crossover from a Bose–Einstein condensate (BEC) of tightly bound bosonic pairs to a Bardeen–Cooper–Schrieffer (BCS) superfluid of long-range Cooper pairs in three dimensions [4–11]. Restricting these gases to two dimensions strongly alters pairing and superfluidity [12–20], and may offer insight into unconventional forms of superconductivity encountered in solid-state physics [21, 22].

    Very recently, S. Jochim's group at Heidelberg University have experimentally probed how the key features of Fermi superfluidity emerge at the most fundamental level — 'from the bottom up' [23, 24]. The group deterministically prepared nearly pure quantum ground states for up to twenty ultracold fermions that were equally distributed between two different spin states and confined in a (quasi-)two-dimensional harmonic trap. Their flexible experimental set-up enabled them to tune the inter-spin interactions from the non-interacting limit into the regime of strong binding, and to extract the single particle and spin resolved momentum distribution of the Fermi gas at any intermediate interaction strength. They reported Cooper pairing in a system comprising only twelve interacting particles, which manifested as a peak in the correlations between atoms with opposing spins and momenta at the Fermi surface in momentum space [24]. In another experiment involving as few as six particles, they observed a few-body precursor of a quantum phase transition from a normal fluid to a superfluid [23]. The precursor transition was signalled by a softening (i.e., a decrease in frequency) of the lowest mode in the excitation spectrum when the attractive interaction strength was increased. In the many-body limit, this mode becomes associated with amplitude variations of the superfluid order parameter and is commonly referred to as the massive 'Higgs mode' [25]. While mode

softening in the six-atom system had previously been predicted [26], to our knowledge, the pair momentum correlations mentioned above have not yet been theoretically calculated.

Earlier theoretical work on two-dimensional trapped mesoscopic Fermi gases has been focused on probing their excitations. In 2016, G. Bruun et al. [26] calculated the monopole (zero angular momentum) excitation spectra for between six and twelve fermions interacting via a contact potential. For closed-shell configurations, they found that the lowest energy mode depends non-monotonically on the interaction strength and mainly consists of coherent excitations of time-reversed pairs — which, as mentioned above, has since been confirmed by experiment [23]. Their approach employed the harmonic oscillator basis, which is convenient for evaluating the Hamiltonian matrix elements, however is poor at approximating the cusps in the wave function induced by the short-range interactions [27]. This made it necessary to use very large numbers of basis states (on the order of $\sim 10^7$) to numerically converge the energies [26], and the size of the calculation made it difficult to solve for two-body observables such as momentum-space pair correlations. More recently in 2022, J. Hofmann et al. [28] approximated the excitation spectra of the same Fermi systems by using an exactly solvable (integrable) $s$-wave pairing Hamiltonian known as the Richardson model [29,30]. While a full contact interaction can couple opposite spins in any combination of harmonic oscillator states, the Richardson model only accounts for time-reversed pairing in the same energy level (or shell) and assumes a constant coupling strength for all pairs. As such, the formalism retains the key matrix elements that give rise to superfluidity [31] and allowed the lowest pair excitation mode to be approximated for the first fifteen closed-shell configurations [28]. It was hence demonstrated how the minimum energy of pair excitations deepens with increasing particle number and shifts toward weaker interaction strengths, consistent with experiment [23].

In this manuscript, we adopt an entirely different and highly accurate (virtually exact) approach for calculating the energetics of two-dimensional trapped mesoscopic Fermi gases, which additionally allows us to determine their structural properties and pair correlation functions. We obtain the excitation spectra variationally, based on the now renowned technique introduced by K. Varga and Y. Suzuki in 1995 [32,33]. The trial wave functions are chosen to be combinations of explicitly correlated Gaussians, which permit an analytical evaluation of the Hamiltonian matrix elements [34,35]. The non-linear variational parameters of these trial functions, the Gaussian widths, are selected stochastically. The suitability of this method to describe ultracold few-particle systems is three-fold [36–38]: 1) Cold atoms are sufficiently dilute that only binary interactions are important. Since each Gaussian basis function depends explicitly on every two-body correlation (interparticle separation) in the system, a very high accuracy is achievable. 2) Cold atoms have universal properties that are independent of the microscopic details of the true interaction potential, justifying the assumption of a Gaussian interaction. 3) The Gaussian basis functions are flexible enough to simultaneously replicate correlations that develop on *any* length scale, including those of the scattering potential and the external confinement. This is because a wave function in a harmonic trap has a naturally Gaussian dependence at large distances, whereas its short-range cusp is well captured by superpositions of Gaussians. Consequently, such an approach has previously been used to obtain numerically exact energies and structural properties (such as radial one-body densities and pair distribution functions, but not pair momentum correlations) for spin-balanced two-component Fermi gases subject to an isotropic three-dimensional harmonic confinement. In 2011, the three-dimensional system was solved for up to six particles at a full range of interaction strengths [39], while subsequently in 2014 and 2015, the eight- [40] and ten-particle [41] problems were also solved at unitarity. For all three atom numbers, pairing could be evidenced by the clear two-peak structure of the (scaled) radial pair distribution functions.

In the two-dimensional calculations reported here, we employ a shape-resonant Gaussian interaction potential — which has a large and variable effective range — to mimic and probe

the *quasi*-two-dimensional nature of real experimental confinement geometries [42–45]. We are able to access the second-order pair correlations measured in experiment [24] by evaluating the matrix elements of the real-space one- and two-body fermionic density matrices in the correlated Gaussian basis and then analytically Fourier transforming the results into momentum space. We focus on studying the correlations in the ground state for spin-balanced two-component Fermi gases in different interaction regimes. The one distinction between our theoretical analysis and the experiment is the number of particles. Whereas the latter involved twelve atoms, the maximum number that we can consider is six due to computational time constraints which are imposed by the first-quantised formulation of the explicitly correlated Gaussian (ECG) method. Nevertheless, our calculation of the pair correlation function is new and our findings complement the experiment in revealing how pairing emerges in the limit of very few fermions.

This paper is organised as follows: In Sec. 2 we discuss our model of the two-dimensional Fermi gas, including the special role played by the effective range of interactions. (Since the ECG method has already been thoroughly detailed in the literature, we distill the essential aspects which apply to solving the system of interest in Appendix A.) In Sec. 3 we present and interpret our results: First, we study the excitation spectrum of the Fermi gas, focusing on the unique behaviour of the lowest monopole mode. Subsequently, we elucidate the nature of opposite-spin pair correlations in the ground state and we directly compare our calculations to experiment. We investigate the effects of particle number, interaction strength, and axial confinement strength on both the excitations and pairing. We conclude and identify avenues for future research in Sec. 4.

## 2 Model

We theoretically consider equal-mass two-component Fermi gases comprising $N = N_\uparrow + N_\downarrow$ atoms with balanced spin populations [i.e., $N_\uparrow = N_\downarrow = N/2$, where $N_\uparrow$ ($N_\downarrow$) is the number of 'spin-up' ('spin-down') fermions]. Such a system is exemplified by ultracold fermionic atoms of $^6$Li prepared in the two lowest $^2S_{1/2}$ hyperfine levels. In the experiments of interest, these particles are confined to a highly anisotropic single layer of a standing-wave optical dipole trap, which freezes out motion along the axial ($z$) direction. This layer is then superimposed with an optical tweezer — or 'microtrap' — which provides an isotropic radial harmonic confinement $\omega_r$ [23, 24, 46]. When superimposed on a large ensemble of atoms, the small microtrap can locally enhance the chemical potential by a significant amount without modifying the temperature of the gas [47]. This leads to a small region of increased densities deep in the degenerate regime, and due to Fermi–Dirac statistics, all low-lying energy levels of the microtrap become filled with almost unit probability [46]. By inclining and lowering the trap walls in a controlled manner, particles above a certain 'spill threshold' can then be deterministically removed, leaving behind a stable *mesoscopic* number of atoms in the ground state [46]. The systems of particular relevance to the current study contain as few as $N_\uparrow + N_\downarrow = 1 + 1$, $2 + 2$, or $3 + 3$ particles, such that in the non-interacting ground state only the first two 2D harmonic oscillator shells are occupied. Interactions (collisions) subsequently induced by a Feshbach resonance between distinguishable fermions in the gas (i.e., between the different hyperfine states) are low in energy and well described by *s*-wave two-body physics.

The system Hamiltonian in two dimensions reads as follows:

$$\mathcal{H} = \sum_{i=1}^{N}\left[-\frac{\hbar^2}{2m}\nabla_{\mathbf{r}_i}^2 + V_{\text{ext}}(|\mathbf{r}_i|)\right] + \sum_{i<j}^{N} V_{\text{int}}(|\mathbf{r}_i - \mathbf{r}_j|)\,, \tag{1}$$

where $m$ is the atomic mass and $\mathbf{r}_i$ denotes the position vector of the $i^{th}$ atom measured from the trap centre. The first term corresponds to the kinetic energy of the particles, the second term to an external harmonic trap,

$$V_{\text{ext}}(|\mathbf{r}_i|) = \frac{m\omega_r^2}{2}\, r_i^2, \quad r_i \equiv |\mathbf{r}_i|\,, \tag{2}$$

and the third term to short-range pairwise interactions between fermions with unlike spins. We model these interactions with a finite-range Gaussian potential [45] that is parameterised by a width $r_0$ ($> 0$) and a depth $V_0$ ($< 0$):

$$V_{\text{int}}(|\mathbf{r}|) = V_0 \exp\left(-\frac{r^2}{2r_0^2}\right) - V_0 \frac{r}{l_r} \exp\left[-\frac{r^2}{2(2r_0)^2}\right], \tag{3}$$

where $l_r = \sqrt{\hbar/(m\omega_r)}$ is the radial harmonic oscillator length scale in the 2D plane. In the non-interacting limit of $V_0 = 0$, the Hamiltonian $\mathcal{H}$ in Eq. (1) has eigenvalues of $\varepsilon^{(0)} = (2n + |m| + 1)\hbar\omega_r$, where $n = 0, 1, 2, \dots$ is the principal quantum number and $m = 0, \pm1, \pm2, \dots$ is the quantum number for orbital angular momentum.

For a fixed value of $r_0$, the value of $V_0$ can be adjusted to generate potentials with different free-space $s$-wave scattering lengths and effective ranges (or equivalently, we may fix $V_0$ and vary $r_0$). We consider two particles elastically scattering via the interaction potential, Eq. (3), in two-dimensional free space. We solve the $s$-wave radial Schrödinger equation for the relative motion up to a radius much larger than $r_0$, matching the logarithmic derivatives of the wave functions to the asymptotic form in order to obtain the real-valued $s$-wave scattering phase shift $\delta(k)$ [48]. Subsequently, by fitting the phase shift to its low-energy expansion in two dimensions,

$$\cot[\delta(k)] = \frac{2}{\pi}\left[\gamma + \ln\left(\frac{k a_{2D}}{2}\right)\right] + \frac{1}{\pi}k^2 r_{2D} + \dots\,, \tag{4}$$

we determine both the $s$-wave scattering length $a_{2D}$ and the effective range $r_{2D}$ [49–51].[1] Here, $k \equiv |\mathbf{k}|$ is the magnitude of the relative wave vector between the two atoms in the 2D plane and $\gamma \simeq 0.577216$ is Euler's constant. At low energy, the physics is independent of the short-range details of the interaction potential and instead exhibits universality with respect to both $a_{2D}$ and $r_{2D}$. Accordingly, in our calculations we choose Gaussian widths small enough, $r_0 \lesssim 0.1 l_r$, to ensure that higher order expansion terms in Eq. (4) are negligible within the energy range of interest. We have furthermore implemented a modified version of the model potential — given by Eq. (S23) in the supplemental material of Ref. [45] — and have found that it yields the same energies as in Fig. 2 for a given two-body binding energy (defined below) and $r_{2D}$. This confirms that effects beyond those of the effective range are indeed negligible.

In two dimensions the scattering length is always positive, $a_{2D} > 0$. In a many-body picture, the two-component Fermi gas undergoes a crossover from a Bose–Einstein condensate of diatomic molecules to a Bardeen–Cooper–Schrieffer superfluid of Cooper pairs as $a_{2D}$ increases. However, unlike in three dimensions, there is no unitary limit where the system becomes scale invariant and the interaction strength (scattering length) diverges. Rather, the strongly interacting regime is in the vicinity of $\ln(k_F a_{2D}) = 0$, where the Fermi wave vector $k_F$ denotes the radius of the non-interacting Fermi sea at zero temperature [52].

---

[1]Note that the precise definitions of the two-dimensional scattering length $a_{2D}$ and the two-dimensional effective range $r_{2D}$ vary in the literature. Our particular definition of $r_{2D}$ has units of squared length, consistent with Ref. [45].

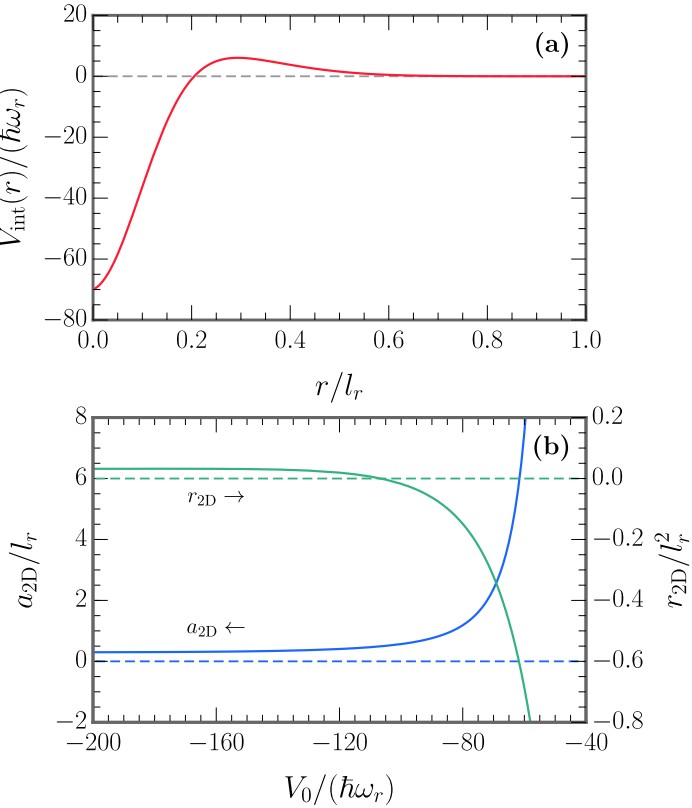

Figure 1: **(a)** The model Gaussian interaction potential, Eq. (3), at $V_0/(\hbar\omega_r) = -70$ and $r_0/l_r = 0.1$ [where $l_r^2 = \hbar/(m\omega_r)$]. **(b)** The two-dimensional scattering length $a_{2D}$ (in blue) and the two-dimensional effective range $r_{2D}$ (in green) as functions of the potential depth $V_0$, for a fixed width of $r_0 = 0.1 l_r$. (Note, this figure is similar to Fig. 1 in Ref. [45].)

As previously described, in cold-atom experiments a two-dimensional geometry can be realised by applying a strong harmonic confinement along the axial direction, with angular frequency $\omega_z$ and length scale $l_z = \sqrt{\hbar/(m\omega_z)}$. Realistically, however, the extent of the gas perpendicular to the 2D plane is necessarily finite. At low energy and small $l_z$ (such that $kl_z \ll 1$), the two-body scattering of distinguishable fermions can be mapped onto a 2D scattering amplitude with an effective range given by [42–45][2]

$$r_{2D} = -l_z^2 \ln(2) . \tag{5}$$

By assigning an appropriately finite and negative value to the effective range parameter in the purely two-dimensional model considered here, we can thus mimic the effect on the scattering of a *quasi*-two-dimensional confining potential. In particular, through our choice of the interaction parameters $V_0$ and $r_0$, we can attribute a value to the dimensionless effective range $r_{2D}/l_r^2$ which matches the trap aspect ratio $\omega_z/\omega_r$ in a given experiment.

In practical computations, we tune the effective range to non-negligible negative values through a shape resonance [45, 53], which arises due to the general structure of the model potential shown in Eq. (3): the first term creates an attractive well that can support virtual

---

[2]For this mapping to be valid, we furthermore require $l_z$ to be much greater than the van der Waals range of the interactions between atoms — i.e., $r_{vdW} \ll l_z < l_r$ — which is always satisfied experimentally.

bound states, while the second term adds a small repulsive barrier that can couple these virtual bound states to free-space scattering states — as depicted in Fig. 1**(a)**. Figure 1**(b)** illustrates the range of combinations of $a_{2D}$ and $r_{2D}$ that can be obtained by fixing $r_0$ and varying $V_0$. In this figure and in all our calculations, we restrict our attention to the regime where the potential supports a single two-body $s$-wave bound state in two-dimensional free space [45].[3]

To numerically solve the time-independent Schrödinger equation for the Hamiltonian in Eq. (1), we employ the method of explicitly correlated Gaussians. A description of this technique is provided in Appendix A. We parameterise our results in terms of the effective range $r_{2D}$ and the two-body binding energy $\varepsilon_b > 0$, with the latter determined by the following approach. For every set of $V_0$ and $r_0$ values that we use to numerically solve a general $N_\uparrow + N_\downarrow$ problem, we also solve the corresponding $1 + 1$ problem numerically by implementing the correlated Gaussian method. This yields the relative energy of the two-body ground state, $\mathcal{E}_{\text{rel}, 1+1}$ (see Appendix A). The total ground-state energy of one spin-$\uparrow$ particle and one spin-$\downarrow$ particle in the 2D harmonic trap is given by $\mathcal{E}_{1+1} = 2\hbar\omega_r - \varepsilon_b$. Since we know that $\mathcal{E}_{1+1} = \mathcal{E}_{\text{com}, 1+1} + \mathcal{E}_{\text{rel}, 1+1}$ and there are no centre-of-mass excitations in the ground state, $\mathcal{E}_{\text{com}, 1+1} = \hbar\omega_r$, we can then immediately obtain $\varepsilon_b$.

## 3 Results

We apply the method of explicitly correlated Gaussians to obtain numerically optimised and converged basis sets at a wide range of attractive interaction strengths (or binding energies) for the fermionic systems of interest. Upon diagonalising the Hamiltonian, we utilise the eigenvalues to calculate the low-energy excitation spectra of the Fermi gases and the eigenvectors to determine their structural properties. With regard to the latter, we focus on investigating the nature of opposite-spin pair correlations in the ground state and we directly compare our numerics against recent experimental measurements.

### 3.1 Excitation Spectrum

The excitation spectra of the Fermi systems are of fundamental interest since they can reveal signatures of pairing [26] and can be experimentally accessed in two dimensions [23]. Figure 2 displays the lowest energy fermionic excitation spectrum, i.e., the difference $\Delta E = E_{1ES} - E_{GS}$ between the first-excited-state ($E_{1ES}$) and ground-state ($E_{GS}$) energies as a function of the two-body binding energy $\varepsilon_b$. In the upper panel **(a)** we compare our results for $N_\uparrow + N_\downarrow = 1 + 1$, $2 + 2$, and $3 + 3$ fermions at very nearly *zero* effective range (numerically, we set $r_{2D}/l_r^2 = -0.001 \approx 0$), while in the lower panel **(c)** our results for $3 + 3$ fermions are compared at *different* fixed values of the effective range. In the middle panel **(b)**, the ground- and first-excited-state energies used to calculate the excitation energies of panel **(a)** are shown separately as a reference.

The non-interacting ground state at $\varepsilon_b = 0$ can assume one of two configurations depending on the total number of particles $N$: either all of the degenerate single-particle states of the highest energy level of the 2D harmonic oscillator are filled ('closed shell'), or some of the degenerate states remain empty ('open shell'). The $1 + 1$ and $3 + 3$ systems both feature a closed-shell ground state that is non-degenerate, whereas the $2 + 2$ ground state is open-shell. We restrict our consideration to ground states that are characterised by zero total orbital an-

---

[3]At the point where a new bound state enters the potential both $a_{2D}$ and $|r_{2D}|$ positively diverge. As discussed in Ref. [45], the potential does not support a two-body bound state in the limit of $V_0 \to 0$. In two dimensions this is in stark contrast to the case of a potential that is everywhere attractive. Such a potential (even one that is arbitrarily weak) always supports a two-body $s$-wave bound state in free space because the scattering amplitude always features a pole at negative energies [52].

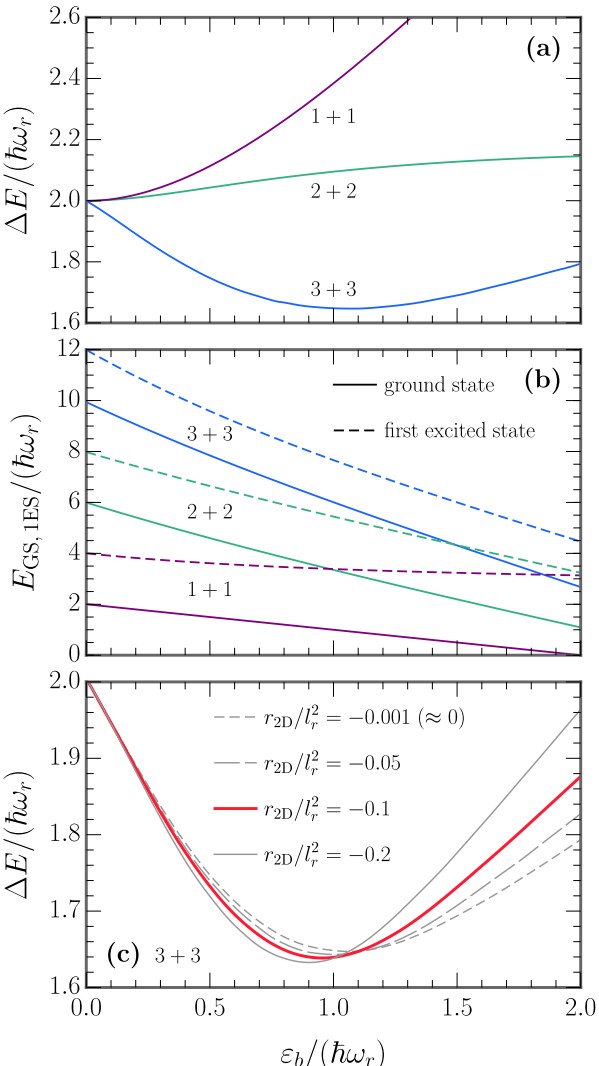

Figure 2: The lowest monopole excitation spectrum for various few-body Fermi systems. **(a)** The excitation energy, $\Delta E = E_{1ES} - E_{GS}$, as a function of the two-body binding energy $\varepsilon_b$ for $N_\uparrow + N_\downarrow = 1+1, 2+2$, and $3+3$ fermions at zero effective range ($r_{2D}/l_r^2 = -0.001 \approx 0$). **(b)** The ground- ($E_{GS}$) and first-excited-state ($E_{1ES}$) energies used to calculate $\Delta E$ of panel **(a)**. Similar to panel **(a)**, the purple, green, and blue lines are associated with the $1+1, 2+2$, and $3+3$ systems, respectively. **(c)** The non-monotonic excitation spectrum for $3+3$ fermions at different effective ranges. The selected values — $r_{2D}/l_r^2 = -0.2, -0.1, -0.05, -0.001$ — respectively correspond to trap aspect ratios of $\omega_r/\omega_z \approx 1/3.5, 1/7, 1/14, 1/700$. [Note that the blue line in **(a)** is the same as the short-dashed gray line in **(c)**.]

gular momentum. For the $2+2$ system, this means that the two highest energy opposite-spin fermions reside in different degenerate single-particle states. Since the Hamiltonian is rotationally symmetric, only monopole excitations between states with the same (i.e., zero) total angular momentum occur.[4] For all three atom numbers at $\varepsilon_b = 0$, the lowest monopole excitation has an energy of $\Delta E = 2\hbar\omega_r$. This can be attributed either to exciting a single particle

---

[4] The $m$ quantum numbers for all atoms sum to zero in both the ground and excited states.

up two harmonic oscillator shells, or to exciting a time-reversed pair of particles $(n, m, \uparrow)$ and $(n, -m, \downarrow)$ up one shell each.

As the attractive interaction strength increases from zero, $\varepsilon_b > 0$, the excitation energies for systems with different particle numbers in panel **(a)** evolve very differently. A striking feature is the non-monotonic behaviour of $\Delta E$ for the case of $3+3$ fermions. As first argued in Ref. [26][5] — and later lucidly discussed in M. Holten's PhD thesis [46] — this non-monotonicity is indicative of *pair correlations*. The first excited state for $3+3$ fermions is a linear combination of three degenerate configurations: one being the result of a single-particle excitation and the other two the result of pair excitations. The energy of the former grows with $\varepsilon_b$ simply because increasing the mean-field attraction felt by each particle enhances the effective confinement, $\omega_r^{\mathrm{eff}} > \omega_r$ — which thereby raises the cost of exciting a single particle, $\Delta E = 2\hbar\omega_r^{\mathrm{eff}}$ [26]. On the other hand, when a pair of particles is excited from the closed-shell ground state they can use the degenerate states in the new, otherwise empty harmonic oscillator level to increase their wave function overlap. This causes them to gain binding energy, and hence, diminishes the cost of monopole excitations monotonically as $\varepsilon_b$ increases [26, 46].[6] At a critical binding energy (denoted by $\varepsilon_b^c$) the excitation energy $\Delta E$ reaches a minimum. Beyond this point the interaction strength becomes comparable to the radial trapping frequency $\varepsilon_b \sim \hbar\omega_r$, which signifies that pairing then occurs not only in the excited states, but also in the ground state. As a result, the ground-state energy starts decreasing faster than the first-excited-state energy, such that $\Delta E$ begins to increase [23]. These pairing effects are dominant in the $3+3$ system which leads to the overall non-monotonic dependence of $\Delta E$ on $\varepsilon_b$. This is not the case for $1+1$ and $2+2$ fermions in the monopole sector, and consequently, for those systems $\Delta E$ increases monotonically with $\varepsilon_b$ instead. In Appendix B, we discuss how our results based on the Gaussian interaction potential of Eq. (3) compare quantitatively to the contact interaction results from Ref. [26].

We can consider approaching the many-body limit by increasing the number of particles $N \to \infty$, while keeping the trap strength $\omega_r$ the same.[7] In this case, if the ground state has a closed-shell configuration, then the minimum value of $\Delta E$ at the critical binding energy $\varepsilon_b^c$ will decrease as $N$ increases, eventually reducing to zero in the many-body limit so that pairs are coherently excited without any energy cost [25, 26]. In this limit, if $\varepsilon_b$ is increased from zero to $\varepsilon_b^c$, then the many-body two-component Fermi gas will become unstable and undergo a second-order phase transition into a superfluid state. The lowest energy monopole excitation of the trapped superfluid corresponds to the Higgs mode with an energy equal to twice the superfluid gap [25, 46]. Our result for $3+3$ fermions in panel **(a)** can thus be viewed as a *few-body precursor* to the Higgs mode for the Gaussian interaction potential given by Eq. (3). In panel **(c)**, we investigate the effect of different quasi-two-dimensional harmonic confinements on this 'few-body Higgs mode' by varying the effective range parameter $r_{\mathrm{2D}}$ introduced in Eq. (4). We plot the lowest monopole excitation energy for $3+3$ fermions at the following

---

[5]This work calculated the monopole excitation spectrum of the same system (but for contact interactions) by using exact diagonalisation in the harmonic oscillator basis.

[6]Similarly, the remaining pairs of particles in the lower harmonic oscillator shell can increase their wave function overlap and gain binding energy by occupying the degenerate states that are now free. Thus, the pair excitation energy is a many-particle quantity that can only be accurately determined by taking the entire mesoscopic sample into account [46].

[7]In free space where BCS theory applies, the many-body limit is typically approached by increasing both the number of particles $N \to \infty$ and the system volume $\mathcal{V} \to \infty$ in such a way that the density $n = N/\mathcal{V}$ remains constant. In our scenario where $\omega_r$ is fixed, we instead have $n \to \infty$ when $N \to \infty$ and pairing is suppressed at small enough binding energies $\varepsilon_b \ll \hbar\omega_r$. If we wish to make our system amenable to BCS theory, we could keep $n$ constant by reducing the trap strength $\omega_r$ while increasing $N$. In that case, the trapping frequency would vanish $\omega_r \to 0$ for $N \to \infty$, which means the condition $\varepsilon_b \ll \hbar\omega_r$ would never be satisfied in the many-body limit. Consequently, a superfluid state with a finite gap would always exist at zero temperature for any non-vanishing interaction strength.

effective ranges: $r_{2D}/l_r^2 = -0.2, -0.1, -0.05, -0.001$ — which are associated with trap aspect ratios of: $\omega_r/\omega_z \approx 1/3.5, 1/7, 1/14, 1/700$, respectively, according to Eq. (5). Notably, we find that as the magnitude of the negative effective range increases, the minimum value of $\Delta E$ decreases and shifts to smaller binding energies, i.e., $\varepsilon_b^c$ decreases. In addition, we see that the dependence of $\Delta E$ on the value of $r_{2D}$ (or $l_z$) is lessened at smaller $\varepsilon_b$. The experiment against which we will later compare our calculated pair momentum correlations had radial and axial trapping frequencies of $\omega_r = 2\pi \times 1{,}101$ Hz and $\omega_z = 2\pi \times 7{,}432$ Hz [24]. These frequencies correspond to a trap aspect ratio of $\sim 1/7$ and an effective range of $r_{2D}/l_r^2 = -0.1027 \approx -0.1$ — designated by the thick red line in panel **(c)**. The value of the critical binding energy for this line is $\varepsilon_b^c \approx 0.953\hbar\omega_r$.

## 3.2 Pair Correlation Function

Pairing — regardless of the exact mechanism by which the particles attract each other — is a correlation phenomenon. This means that we can extract its description from the quantum two-body density matrix, which contains a complete set of information on all two-body correlations in the system [54]. In the position representation, the two-body density matrix reads as follows:

$$\rho(\mathbf{r}_1, \mathbf{r}_1'; \mathbf{r}_2, \mathbf{r}_2') = \langle \psi_\uparrow^\dagger(\mathbf{r}_1)\psi_\uparrow(\mathbf{r}_1')\psi_\downarrow^\dagger(\mathbf{r}_2)\psi_\downarrow(\mathbf{r}_2') \rangle, \tag{6}$$

where $\langle \cdots \rangle$ denotes an expectation value, and $\psi_\sigma^\dagger(\mathbf{r})$ and $\psi_\sigma(\mathbf{r})$ are fermionic field creation and annihilation operators, respectively (with $\sigma = \uparrow, \downarrow$). The diagonal matrix elements of Eq. (6) correspond to the instantaneous correlations between all particles' positions, whereas the off-diagonal elements are responsible for two-particle coherence [54]. The diagonal elements are of particular interest since they are directly accessible in experiments. These elements, $\langle \eta_\uparrow(\mathbf{r}_1)\eta_\downarrow(\mathbf{r}_2) \rangle$ — written using the density operator, $\eta_\sigma(\mathbf{r}) = \psi_\sigma^\dagger(\mathbf{r})\psi_\sigma(\mathbf{r})$ — specifically provide the probability of simultaneously finding opposite-spin fermions at positions $\mathbf{r}_1$ and $\mathbf{r}_2$. They can equivalently be written as $\langle n_\uparrow(\mathbf{p}_1)n_\downarrow(\mathbf{p}_2) \rangle$ — with $n_\sigma(\mathbf{p} = \hbar\mathbf{k})$ the momentum-space density operator — in order to give the probability of simultaneously finding opposite-spin fermions with momenta $\mathbf{p}_1$ and $\mathbf{p}_2$. Since the signatures of opposite-spin pairing are predominantly evident in the momentum correlations, we focus on the latter. Note that even in the purely non-interacting regime, coincidences of a spin-$\uparrow$ fermion with momentum $\mathbf{p}_1$ and a spin-$\downarrow$ fermion with momentum $\mathbf{p}_2$ can still occur. In this limit, the two-particle density distribution becomes a direct product of independent single-particle densities: $\langle n_\uparrow(\mathbf{p}_1)n_\downarrow(\mathbf{p}_2) \rangle = \langle n_\uparrow(\mathbf{p}_1) \rangle \langle n_\downarrow(\mathbf{p}_2) \rangle$ [54]. We therefore subtract this quantity so as to only account for correlations caused solely by interactions, leading to the second-order correlation function, $\mathcal{C}^{(2)}$, that features in S. Jochim's experiments [24]:

$$\mathcal{C}^{(2)}(\mathbf{p}_1, \mathbf{p}_2) = \langle n_\uparrow(\mathbf{p}_1)n_\downarrow(\mathbf{p}_2) \rangle - \langle n_\uparrow(\mathbf{p}_1) \rangle \langle n_\downarrow(\mathbf{p}_2) \rangle. \tag{7}$$

We theoretically evaluate $\mathcal{C}^{(2)}$ by using the method of explicitly correlated Gaussians, relegating the details of this calculation to the appendices, while focusing the main text on a discussion of our results. In Appendix C, we define the expectation values in Eq. (7) in terms of the one- and two-body fermionic density matrices in position and momentum space. The real-space one-body density matrix in the correlated Gaussian basis has previously been derived in Ref. [39] for the case of an isotropic three-dimensional harmonic confinement. In Appendix D, we perform the analogous derivation in two dimensions and then analytically Fourier transform the result to determine expressions for $\langle n_\uparrow(\mathbf{p}_1) \rangle$ and $\langle n_\downarrow(\mathbf{p}_2) \rangle$. In Appendix E, we extend this approach to obtain the correlated Gaussian matrix elements of the real-space two-body density matrix. The Fourier transformation into momentum space can again be carried out analytically to yield an expression for $\langle n_\uparrow(\mathbf{p}_1)n_\downarrow(\mathbf{p}_2) \rangle$.

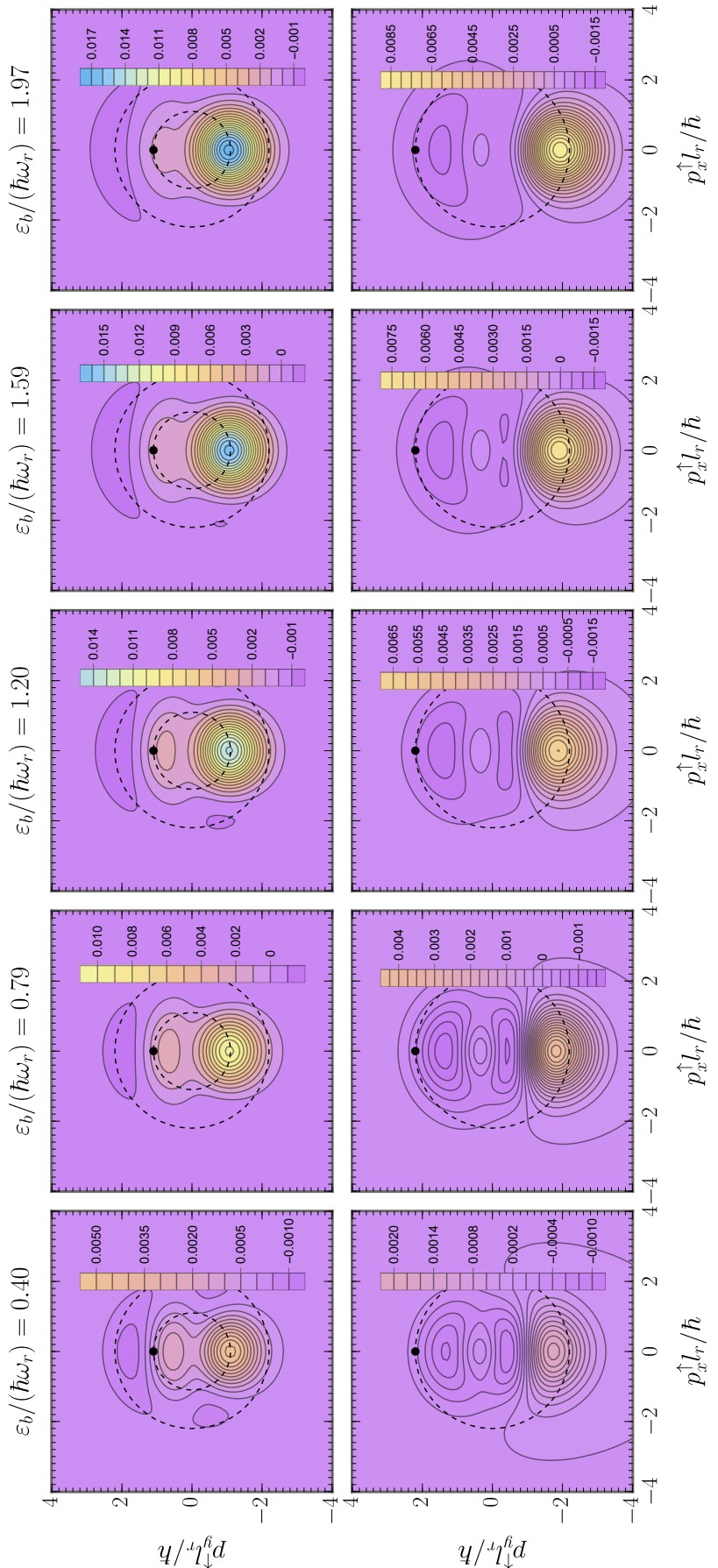

Figure 3: $\mathcal{C}^{(2)}(\mathbf{p}_\uparrow, \bar{\mathbf{p}}_\downarrow)$ as a function of $\mathbf{p}_\uparrow$ with $\bar{\mathbf{p}}_\downarrow$ fixed at the black point for $3+3$ fermions in the ground state. The radii of the dashed circles signify $|\bar{\mathbf{p}}_\downarrow|$ and $p_F$.

320     A pertinent question is how to define (or approximate) the Fermi momentum $p_F = \hbar k_F$
321 in a few-body system. The harmonic trap in the radial direction provides not only a natural
322 length scale for the Fermi gas $l_r = \sqrt{\hbar/(m\omega_r)}$, which sets the average interparticle spacing,
323 but also a natural momentum scale $p_r = \sqrt{\hbar m\omega_r}$. When there are only very few particles, the
324 step in the momentum distribution at $p_F$ for a given spin component is 'smeared out', with a
325 width on the order of $p_r$. Thus, while the mesoscopic sample is characterised by two distinct
326 momentum scales $p_r$ and $p_F$, an unambiguous definition of $p_F$ does not exist because the Fermi
327 surface is coarse-grained [46]. One option in this case is to simply use the continuum equation
328 which typically defines the Fermi momentum in a many-body system $p_F = \sqrt{2m\varepsilon_F}$, where the
329 Fermi energy $\varepsilon_F$ is the energy of the non-interacting ground state at zero temperature. Instead,
330 we choose to define $p_F$ in a manner consistent with Ref. [55] which also theoretically probes
331 the many-body physics of two-component Fermi gases from the few-body regime. Therein the
332 authors employ the local density approximation (LDA) in three dimensions to determine $p_F$
333 as a smooth function of the number of particles $N \leq 10$. Although the applicability of either
334 the continuum equation or the LDA to such small atom numbers may be questioned, the latter
335 approach minimises few-body shell effects and smoothly extrapolates to the correct result in
336 the large-$N$ limit. We therefore define a local chemical potential $\mu(\mathbf{r}) = \mu - V_{\text{ext}}(|\mathbf{r}|)$, which
337 depends on the global chemical potential $\mu = \partial\varepsilon/\partial N$, where $\varepsilon$ is the total energy of the trapped
338 gas. In two dimensions, a trapped non-interacting Fermi gas with balanced spin populations
339 has the particle number density,

$$n(\mathbf{r}) = \frac{m}{\pi}\left(\mu - \frac{m\omega_r^2}{2}r^2\right), \tag{8}$$

340 which gives the total number of particles,

$$N = 2N_\uparrow = \int d^2\mathbf{r}\, n(\mathbf{r}) = \frac{\mu^2}{\omega^2}. \tag{9}$$

341 Above, the radial co-ordinate $r \equiv |\mathbf{r}|$ is integrated from zero up to the Thomas–Fermi radius
342 $r_{\text{TF}} = \sqrt{2\mu/(m\omega_r^2)}$. By using the definition of the trap length $l_r$, we then immediately obtain

$$p_F = (8N_\uparrow)^{1/4}\hbar/l_r \tag{10}$$

343 as the local Fermi momentum at the centre of the trap.

344     We first take the correlation function $\mathcal{C}^{(2)}(\mathbf{p}_1, \mathbf{p}_2)$ in Eq. (7) and fix $\mathbf{p}_2$ to a single value
345 denoted by $\bar{\mathbf{p}}_2$, while allowing $\mathbf{p}_1$ to vary. We plot the results for the ground state of the $N_\uparrow +$
346 $N_\downarrow = 3 + 3$ Fermi system in Fig. 3. The effective range is set to the experimental value in all
347 panels, $r_{2D}/l_r^2 = -0.1$, and the binding energy increases across the panels from left to right.
348 We consider all (non-zero) binding energies measured in Fig. 2 of Ref. [24]: $\varepsilon_b/(\hbar\omega_r) =$
349 0.79, 1.20, 1.97 — except for $\varepsilon_b/(\hbar\omega_r) = 15.90$[8] — and two additional intermediate values:
350 $\varepsilon_b/(\hbar\omega_r) = 0.40, 1.59$. The horizontal and vertical axes on each plot respectively measure
351 the $x$ and $y$ components of $\mathbf{p}_1 \equiv \mathbf{p}_\uparrow$. The value of $\bar{\mathbf{p}}_2 \equiv \bar{\mathbf{p}}_\downarrow$ is indicated by the black point ($\bullet$)
352 and a dashed circle is drawn at that radius, while another dashed circle is drawn at the radius
353 of the Fermi momentum $p_F$. In the upper panels $\bar{\mathbf{p}}_2$ is located inside the Fermi sea, whereas
354 in the lower panels it is positioned on the Fermi surface. All panels utilise the same colour
355 scaling. Our figure can be directly compared against plots (a)–(j) in Fig. 2 of Ref. [24]. As was

---

[8]At this binding energy, the 6+6 system in the experiment formed bosonic pairs that were strongly interacting [24]. In the BEC limit of even higher binding energies, the particles would form non-interacting point-like molecules that reside in the ground state of the harmonic oscillator [11]. Later in Sec. 3.3 where we determine the number of pairs in the $3 + 3$ ground state, we will find that we are never close to the deep BEC regime for our considered range of binding energies, $\varepsilon_b \lesssim 2\hbar\omega_r$.

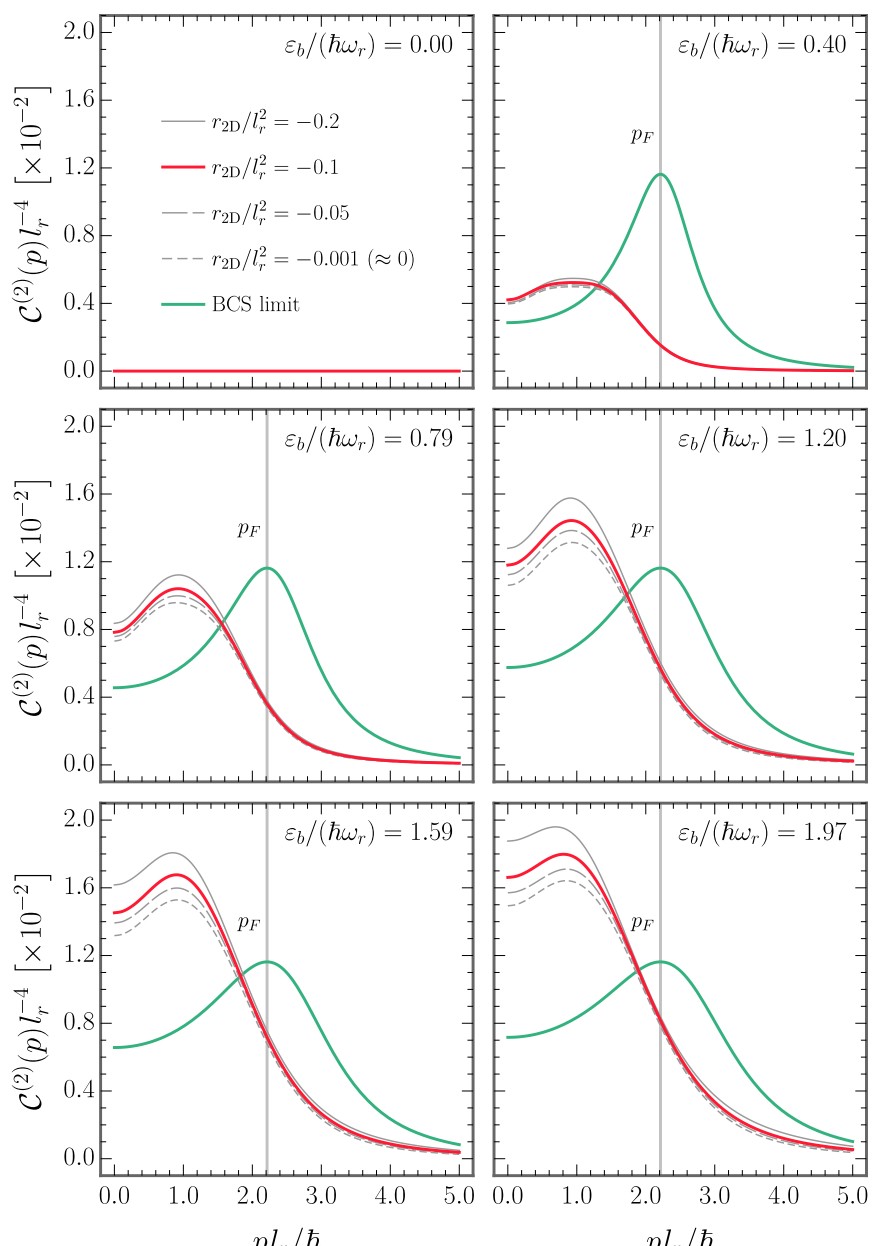

Figure 4: The calculated opposite-momentum pair correlator $\mathcal{C}^{(2)}(p)$ as a function of momentum $p$ for the ground state of $N_\uparrow + N_\downarrow = 3 + 3$ fermions. The multiple panels are associated with different interaction strengths $\sim \varepsilon_b$, whereas the axial confinement $\sim r_{2D}$ is varied within each panel. In the experiment of Ref. [24] the measured binding energies were $\varepsilon_b/(\hbar\omega_r) = 0.00, 0.79, 1.20$, and $1.97$, while the trap aspect ratio corresponded to an effective range of $r_{2D}/l_r^2 = -0.1$ (marked by the thick red line). The vertical gray line designates the Fermi momentum $p_F$.

found in experiment, for particles with different spins there are only considerable second-order correlations between those which have opposing momenta. However, in contrast to the experiment we see that pairing in the 3+3 system is dominant inside the Fermi sea, rather than on the Fermi surface, at all considered binding energies. The experiment for $6 + 6$ fermions instead showed pairing to be dominant on the Fermi surface at binding energies of $\varepsilon_b/(\hbar\omega_r)$

361 $= 0.79$, 1.20, and 1.97.

362     In view of Fig. 3, we define the opposite-momentum pair correlator as $\mathcal{C}^{(2)}(\mathbf{p}_1 = \mathbf{p}, \mathbf{p}_2 =$
363 $-\mathbf{p})$, as was done in Ref. [24]. Due to radial symmetry, $\mathcal{C}^{(2)}(\mathbf{p}, -\mathbf{p}) \equiv \mathcal{C}^{(2)}(p)$ only depends
364 on the magnitude of the particles' momenta and can thus be expressed as a one-dimensional
365 correlation function. $\mathcal{C}^{(2)}(p)$ is plotted in its dependence on momentum $p$ for $3 + 3$ fermions
366 in the ground state in Fig. 4. We explore the parameter space by varying both the two-body
367 binding energy $\varepsilon_b$ and the effective range $r_{2D}$. Each panel is associated with one of the bind-
368 ing energies previously considered in Fig. 3. Inside a given panel, the thick red line corre-
369 sponds to the experiment's value of the effective range, $r_{2D}/l_r^2 = -0.1$, whereas the thin gray
370 lines correspond to the other effective ranges featured in the excitation spectra of Fig. 2(c).
371 [Note that in every panel of Fig. 4, there is one point along the red curve that matches with
372 one point in the associated 2D contour plot of Fig. 3 (with the same binding energy) — but
373 otherwise, these figures contain different information.] Similar to in Ref. [24], we include as
374 a green line the limit from standard BCS theory (normalised to the correct number of parti-
375 cles), which is valid when the mean-field superfluid gap greatly exceeds the binding energy:
376 $\Delta = \sqrt{2\varepsilon_b \varepsilon_F} \gg \varepsilon_b$ [12], where $\varepsilon_F = p_F^2/(2m)$ denotes the Fermi energy. While this result
377 is not quantitatively accurate for only six (or twelve) particles because it neglects quantum
378 fluctuations, it nonetheless provides a qualitative picture of the many-body limit — namely, a
379 single peak at the Fermi momentum $p_F$. The details of the BCS calculation can be found in
380 Ref. [24] and are reproduced here in Appendix F for completeness.

381     Across all panels of Fig. 4, we observe that the strength of the correlations (the maximum
382 height of the peak) increases with increasing binding energy. This aligns with expectations
383 that larger binding energies (or interaction strengths) lead to an increase in pairing. On the
384 other hand, the horizontal position of the peak's maximum barely changes with the binding
385 energy. Within a panel, we see that increasing the magnitude of the negative effective range
386 (at a fixed binding energy) also enhances the pair correlations. But again, this does not shift
387 the peak horizontally. We therefore conclude that opposite spin and momentum pairing for
388 $3 + 3$ fermions is consistently largest at momenta significantly below the Fermi surface. This
389 again contrasts with the experimental measurements for $6 + 6$ fermions [24], where $\mathcal{C}^{(2)}(p)$
390 was observed to peak at $p = p_F$ for the same range of binding energies, $\varepsilon_b \lesssim 2\hbar\omega_r$.

391     In Fig. 5, we overlay the theoretical results on the experimental measurements mentioned
392 above at binding energies of $\varepsilon_b/(\hbar\omega_r) = 0.79$, 1.20, and 1.97 — taken from plots (l), (m),
393 and (n) in Fig. 2 of Ref. [24]. In each panel the smooth red, blue, and green curves show the
394 calculated opposite-momentum pair correlator $\mathcal{C}^{(2)}(p)$ as a function of momentum $p$ for the
395 ground state of $1+1$, $2+2$, and $3+3$ fermions, respectively (with $r_{2D}/l_r^2 = -0.1$). The purple
396 line is the experimental data for the $6 + 6$ ground state. To properly compare systems with
397 different particle numbers we rescale the momentum along the horizontal axis by the Fermi
398 momentum $p_F$. [Note that our definition of the Fermi momentum, Eq. (10), differs slightly
399 from the continuum definition used in Ref. [24]. For 6+6 (3+3) fermions this difference is only
400 about 7% (10%).] Due to the rescaling, we can see that qualitatively — and quantitatively at
401 large momenta, $p > p_F$ — there is minimal difference in the pairing between the 2+2 and 3+3
402 systems. This may be related to the fact that the non-interacting ground state for both four and
403 six particles involves the same number of harmonic oscillator shells. Notably, the experimental
404 $\mathcal{C}^{(2)}(p)$ function peaks at $p_F$ and vanishes at $p \to 0$,[9] while the theoretical $\mathcal{C}^{(2)}(p)$ function for
405 fewer particles peaks well below the Fermi surface and remains finite at small momenta. We
406 remark that the depicted $\mathcal{C}^{(2)}(p)$ results for $1 + 1$ fermions have been compared to the results

---

[9]It should be noted that the error bars on the experimental data in Fig. 5 are much larger at small momenta than at high momenta. This is because $\mathcal{C}^{(2)}(p)$ is experimentally determined by 'counting' pairs of atoms with opposite spins and momenta that occur anywhere around a 'ring' of radius $p$ in momentum space, and then dividing by that radius. Due to a purely statistical effect, at very small radii the numbers of counts are also very small, which means those data points are inherently less reliable.

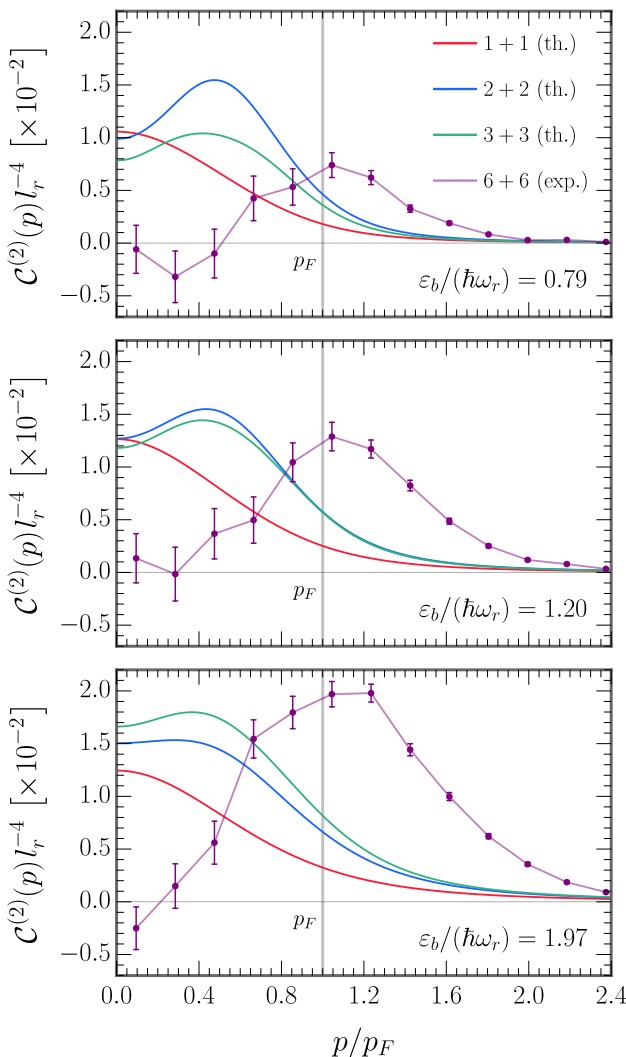

Figure 5: The opposite-momentum pair correlator $\mathcal{C}^{(2)}(p)$, plotted as a function of the rescaled momentum $p/p_F$, and compared for different particle numbers. Each panel corresponds to a different binding energy $\varepsilon_b$. The smooth red, blue, and green curves are the theoretical results for $N_\uparrow + N_\downarrow = 1 + 1, 2 + 2$, and $3 + 3$ fermions in the ground state, respectively (at the experiment's value of the effective range, $r_{2\mathrm{D}}/l_r^2 = -0.1$), while the purple line is the experimental data for the $6 + 6$ fermion ground state. The vertical gray line designates the Fermi momentum $p_F$.

of another method of exact diagonalisation which uses a numerical B–spline basis set,[10] and in all cases, the agreement was found to be exact.

## 3.3 Number of Pairs

We can compute the number of opposite-momentum pairs, $N_{\mathrm{pair}}$, by integrating $\mathcal{C}^{(2)}(p)$ over two-dimensional momentum space. In Fig. 6, we plot $N_{\mathrm{pair}}$ (red points) as a function of interaction strength $\varepsilon_b$ for the 3+3 closed-shell ground state (with $r_{2\mathrm{D}}/l_r^2 = -0.1$). This figure directly

---

[10]B–splines are piece-wise polynomials which can be defined through recursive relations [56]; for a review of their application to quantum atomic and molecular physics, consult Ref. [57].

illustrates how the system evolves from an unpaired to a paired state. For much stronger interactions than those shown, $\varepsilon_b \gg 2\hbar\omega_r$, all the fermions form tightly bound bosonic dimers, reminiscent of the deep BEC regime in macroscopic systems, and the number of pairs becomes maximal, $N_{\text{pair}} = 3$.[11] Here, we see that for weak-to-moderate interactions only a small fraction of the system is paired. The analogous experimental data for the $6+6$ closed-shell ground state is provided in Fig. 3 of Ref. [24]. In a closed-shell structure, all the energy levels up to the Fermi energy are fully occupied and there is a gap of $\hbar\omega_r$ between the completely filled and completely empty levels. This energy gap stabilises the state against small perturbations, and consequently, pairing is suppressed at small binding energies, $\varepsilon_b \ll \hbar\omega_r$ [46]. A critical binding energy $\varepsilon_b^c$ must be reached before it becomes energetically favourable to excite fermions into the empty higher levels and form pairs [46]. As we discussed in the final paragraph of Sec. 3.1, we can approach the many-body limit by increasing the number of particles $N \to \infty$, while keeping the trap strength $\omega_r$ fixed. In this limit, the system remains in the normal state for $\varepsilon_b \ll \hbar\omega_r$ and undergoes a quantum phase transition to a superfluid state with long-range order at $\varepsilon_b^c$. On the mesoscopic scale a precursor of this phase transition can be observed in the fermionic excitation spectra of systems with a closed-shell ground state. The critical binding energy for $3+3$ fermions is associated with the minimum energy of the lowest monopole excitation in Fig. 2(c) — for $r_{\text{2D}}/l_r^2 = -0.1$ (i.e., the thick red line) this value is $\varepsilon_b^c \approx 0.953\hbar\omega_r$. The prediction for $N_{\text{pair}}$ from standard mean-field BCS theory (see either Ref. [24] or Appendix F) is given by the solid blue line in Fig. 6. In order to describe mesoscopic samples, the authors of Ref. [24] off-set the BCS result by the critical binding energy as a type of first-order approximation of finite-size effects. In Fig. 6, we find that the shifted model (dashed green line) fits our numerics (red points) very well for $\hbar\omega_r \lesssim \varepsilon_b \lesssim 2\hbar\omega_r$. Below this however, the grand canonical ensemble on which the model is based leads to a sharp onset of pairing at $\varepsilon_b^c$ [24]. By contrast, we see that the $3+3$ system smoothly transitions into a paired state for $\varepsilon_b > 0$ due to the small fixed particle number. A similar smooth transition was observed for the $6+6$ system [24], corroborating how the ground-state paired fraction evolves with interaction strength in mesoscopic Fermi gases.

## 3.4  Discussion

By comparing the results for the pair correlation function of Sec. 3.2 with those from the experiment in Ref. [24], one could surmise that the transition from an atomic Fermi system with few-body pairing to one with (qualitatively) many-body pairing occurs somewhere between six and twelve particles. We point out that in two dimensions, as was eloquently discussed in Ref. [58], there is a strong connection between the few- and many-body physics of fermion pairing: Elementary quantum mechanics shows that for two isolated particles in a vacuum (such as two distinguishable spin–1/2 fermions), a bound state always exists for an arbitrarily weak, purely attractive interaction [58]. It can also be shown that the existence of a two-body bound state for isolated particles is a necessary and sufficient condition for the Cooper instability of the many-body Fermi sea [12]. This connection is not present in three dimensions: In that case, the interactions must reach a threshold strength before they are able to bind two isolated particles. This means that pairing at arbitrarily weak interactions in three dimensions must be entirely attributed to many-body effects [58]. When the two fermions are on top of a non-interacting filled Fermi sea, rather than in vacuum, the density of available scattering states is altered due to the presence of the other atoms. Momentum states beneath the Fermi surface are unavailable due to Pauli blocking, and at weak interactions, the particles' momenta are restricted to a narrow shell just above the Fermi surface. The three-dimensional density of

---

[11]While our calculations suggest this to be the case, at strong binding energies of $\varepsilon_b > 2\hbar\omega_r$ it is challenging to properly model the tight composite bosonic wave functions, and thus, to obtain *fully* numerically converged energies and structural properties within a reasonable time frame.

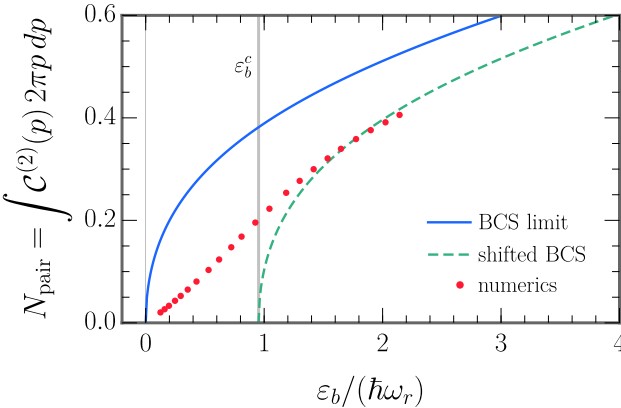

Figure 6: The number of opposite-momentum pairs $N_{\text{pair}}$ (red points) as a function of interaction strength $\varepsilon_b$ for the $3+3$ fermion ground state (with $r_{\text{2D}}/l_r^2 = -0.1$). The maximum possible number of pairs is $N_{\text{pair}} = 3$. At larger binding energies, the mesoscopic sample can be accurately described by shifting the result from standard BCS theory by the critical binding energy, $\varepsilon_b^c \approx 0.953\hbar\omega_r$ (vertical gray line) [24].

states is proportional to the square root of the energy $\rho_{\text{3D}}(\varepsilon) \propto \sqrt{\varepsilon}$, but at the Fermi surface it becomes constant $\rho_{\text{3D}}(\varepsilon_F)$ just like in two dimensions. The effectively reduced dimensionality of the system hence allows the formation of a two-body bound state for arbitrarily weak attraction [58, 59].

In the many-body regime, Cooper pairing tends to be concentrated at the Fermi surface regardless of whether the system is two- or three-dimensional. This is because any two distinguishable particles need to scatter in order to pair (i.e., to become entangled). Likewise, the system needs to build up a superposition of many momenta in order to form a paired state. (This is made clear, for example, by recalling the structure of the ansatz for the superfluid ground-state wave function in standard BCS theory [59, 60].) However, Pauli blocking prevents these processes from happening deep inside the Fermi sea. For the Fermi sea to pair, some scattering states would need to be made available at low momenta — and this would require removing some particles from the Fermi sea by scattering them across a large momentum. The attractive interactions must therefore become strong enough to make it energetically favourable for those particles to scatter out. For weak-to-moderate interactions pairing is hence localised at the Fermi surface due to Pauli blocking, but begins to spread deeper into the Fermi sea as the interaction strength increases [59]. For very strong interactions the Fermi surface completely breaks up and pairing occurs at all momenta. In this limit the many-body system transitions from Cooper pairs to molecules [24, 60].

Having only very few particles thus leads to the question of how strong is the Pauli blocking effect of the Fermi sea? Indeed, the extent of the occurrence of Pauli blocking can be considered a measure of the extent to which the system can be legitimately called a 'Fermi sea'. Because the experimental $\mathcal{C}^{(2)}(p)$ function peaks at the Fermi momentum $p_F$ for a wide range of interaction strengths, $\varepsilon_b \lesssim 2\hbar\omega_r$, this suggests that $6+6$ fermions is already approaching the number of particles required for a quantum many-body system and essentially constitutes a Fermi sea. By contrast, the theoretical $\mathcal{C}^{(2)}(p)$ function peaks substantially below the Fermi momentum in the same interaction regime. This indicates that $3+3$ fermions is still a few-body system in which the low-momentum states are paired. It would therefore be of considerable interest to extend our calculations to $4+4$, $5+5$, and $6+6$ particles to confirm this interpretation. Alternatively, it would be interesting to experimentally measure the pair correlation

function for a number of particles smaller than $6+6$ [24] to compare against our results. As we discuss in Appendix A, the main burden on computational time for increasing particle number is the rapid increase in the number of permutations required to antisymmetrise the wave function — which currently limits our investigation to $3+3$ atoms. If the $6+6$ calculation were feasible timewise, then the additional full harmonic oscillator shell in the non-interacting ground state may be enough to qualitatively modify the outcome from the $3+3$ case. In three dimensions, energies and some structural properties (but not opposite-momentum pair densities) have previously been obtained for $4+4$ [40] and $5+5$ [41] fermions at unitarity by using basis sets that account for the most important but not all correlations. However, this approach may be less accurate at weak-to-moderate interactions. Besides particle number, another factor which may have played a role in the difference of results is temperature, i.e., our calculations assumed zero temperature, while the experiment was performed at a finite temperature which led to a ground-state fidelity of 76% [24]. Nevertheless, we expect this to be less significant since many-body Monte–Carlo simulations have shown that temperature affects the weight and sharpness of the pair correlation peak, rather than shifting the peak to lower or higher momenta [61,62].

# 4 Conclusions and Outlook

In summary, we have used the method of explicitly correlated Gaussians to study the excitations and pairing in two-dimensional trapped mesoscopic Fermi gases. For the closed-shell configuration of $3+3$ fermions, we reproduced the known [23, 26] non-monotonic dependence of the lowest monopole mode on the attractive interaction strength. For $1+1$, $2+2$, and $3+3$ fermions in the ground state, we found that time-reversed pairing is predominant at momenta significantly below the Fermi momentum. We explored the effects of varying the interaction strength (binding energy) and axial confinement (effective range) on the system properties. The difference between the experimental measurements for $6+6$ fermions (where pairing mainly occurred at the Fermi surface) [24] and the calculations for $3+3$ fermions is yet to be resolved. Improving the computational methodology to handle particle numbers greater than six — or conversely, obtaining the experimental data for fewer than twelve particles — would help to fill in this picture.

There are many avenues for future theoretical work on this topic. Means of increasing the numerical convergence rate for stronger binding energies, $\varepsilon_b > 2\hbar\omega_r$, (in addition to higher particle numbers) should continue to be sought. It would moreover be experimentally relevant to compare our (quasi-)two-dimensional calculation to a pure three-dimensional one and to confirm the effect of finite temperature in mesoscopic samples. Another extension would be to consider finite angular momentum sectors which become relevant in the case of anisotropic trapping potentials or spin imbalances. For instance, could one engineer a 'few-body probe' of the Fermi–polaron problem [63]? Finally, in view of the large-scale quench experiments reported in Ref. [64], it would be useful to simulate the effect of an interaction quench in the few-body limit in order to shed further light on the dynamics of the emergence of superfluidity in two-component Fermi gases.

# Acknowledgements

The authors would very much like to thank Jesper Levinsen, Meera Parish, Andy Martin, Alex Kerin, Mitchell Knight, Xia-Ji Liu, and Hui Hu for interesting discussions of the theory; and also Selim Jochim, Marvin Holten, Sandra Brandstetter, and Carl Heintze for helpful discussions

533  about the experiment. We furthermore thank Desmond (Xiangyu) Yin for writing the first it-
534  eration of the C code for Hamiltonian diagonalisation via the method of explicitly correlated
535  Gaussians. This research was supported by the Australian Research Council Centre of Excel-
536  lence in Future Low-Energy Electronics and Technologies, 'FLEET' (Project No. CE170100039),
537  and funded by the Australian government. Emma Laird was supported by a Women–in–FLEET
538  research fellowship.

## A  Method of Explicitly Correlated Gaussians

540  To numerically solve the time-independent Schrödinger equation for the Hamiltonian given by
541  Eq. (1), we complement the stochastic variational method with the use of explicitly correlated
542  Gaussian basis functions [36]. In this section, we provide a concise pedagogical discussion
543  of the main components of this approach which apply to solving systems of trapped two-
544  component fermions. Other works which have implemented this technique in the same context
545  include Refs. [39–41, 65–67].
546      Due to the quadratic form of both the kinetic energy and the external potential energy, the
547  Hamiltonian (1) can be separated into a centre-of-mass component and a relative component:
548  $\mathcal{H} = \mathcal{H}_{\text{com}} + \mathcal{H}_{\text{rel}}$. We define a set of independent Jacobi co-ordinates $\mathbf{x} = (\mathbf{x}_1, \mathbf{x}_2, \ldots, \mathbf{x}_N)^T$,
549  where $\mathbf{x}_N = (\mathbf{r}_1 + \mathbf{r}_2 + \cdots + \mathbf{r}_N)/N$ denotes the centre-of-mass position and $(\mathbf{x}_1, \mathbf{x}_2, \ldots, \mathbf{x}_{N-1})^T$
550  corresponds to relative motion degrees of freedom. The eigenfunctions of the centre-of-mass
551  Hamiltonian are just the well known non-interacting states of the two-dimensional harmonic
552  oscillator for a particle with mass $M = m_1 + m_2 + \cdots + m_N$: $\mathcal{H}_{\text{com}}\Psi_{\text{com}}(\mathbf{x}_N) = \mathcal{E}_{\text{com}}\Psi_{\text{com}}(\mathbf{x}_N)$.
553  Thus, it only remains to solve the Schrödinger equation for the relative motion: $\mathcal{H}_{\text{rel}}\Psi_{\text{rel}}(\mathbf{x}_1,$
554  $\mathbf{x}_2, \ldots, \mathbf{x}_{N-1}) = \mathcal{E}_{\text{rel}}\Psi_{\text{rel}}(\mathbf{x}_1, \mathbf{x}_2, \ldots, \mathbf{x}_{N-1})$ [68].
555      The Jacobi vectors $\mathbf{x}$ and single-particle co-ordinates $\mathbf{y} = (\mathbf{r}_1^\uparrow, \mathbf{r}_2^\downarrow, \mathbf{r}_3^\uparrow, \ldots, \mathbf{r}_N^\downarrow)$ are related by
556  an $N \times N$ linear transformation matrix $\mathbb{U}$ [68]:

$$\mathbf{x} = \mathbb{U}\mathbf{y} \quad \longrightarrow \quad \mathbf{x}_i = \sum_{j=1}^N \mathbb{U}_{ij}\mathbf{r}_j^\sigma, \quad \mathbf{r}_i^\sigma = \sum_{j=1}^N (\mathbb{U}^{-1})_{ij}\mathbf{x}_j \quad (i = 1, \ldots, N). \tag{A.1}$$

557  Here, we have introduced a superscript on the single-particle co-ordinates to designate the
558  pseudospin ($\sigma = \uparrow, \downarrow$) and have ordered them in such a way that the first atom is spin-up, the
559  second is spin-down, the third is spin-up, and so forth. Note, in addition, that $\mathbf{x}$ and $\mathbf{y}$ are
560  'supervectors' (or vectors of vectors) and the double-line font is used in this work to signify a
561  matrix. For two-component Fermi gases with balanced spins ($N = 2N_\uparrow = 2N_\downarrow$), we choose to
562  construct $\mathbb{U}$ in a manner following Ref. [41]: The first $N_\uparrow$ Jacobi co-ordinates correspond to the
563  distances between unlike pairs of fermions. The next $N_\uparrow/2$ [or $(N_\uparrow - 1)/2$ if $N_\uparrow$ is odd] Jacobi
564  co-ordinates correspond to the distances between the centres of mass of the first and second
565  pair, the third and fourth pair, and so on. The remaining Jacobi vectors connect the larger sub-
566  units. For example, in the case of $N = 6$ the transformation matrix is (with $m_{12\cdots i} \equiv m_1 + m_2$
567  $+ \cdots + m_i$ and $m_{12\cdots N} \equiv m_1 + m_2 + \cdots + m_N = M$):

$$\mathbb{U} = \begin{pmatrix} 1 & -1 & 0 & 0 & 0 & 0 \\ 0 & 0 & 1 & -1 & 0 & 0 \\ 0 & 0 & 0 & 0 & 1 & -1 \\ m_1\mathbf{r}_1^\uparrow/m_{12} & m_2\mathbf{r}_2^\downarrow/m_{12} & -m_3\mathbf{r}_3^\uparrow/m_{34} & -m_4\mathbf{r}_4^\downarrow/m_{34} & 0 & 0 \\ m_1\mathbf{r}_1^\uparrow/m_{1234} & m_2\mathbf{r}_2^\downarrow/m_{1234} & m_3\mathbf{r}_3^\uparrow/m_{1234} & m_4\mathbf{r}_4^\downarrow/m_{1234} & -m_5\mathbf{r}_5^\uparrow/m_{56} & -m_6\mathbf{r}_6^\downarrow/m_{56} \\ m_1\mathbf{r}_1^\uparrow/M & m_2\mathbf{r}_2^\downarrow/M & m_3\mathbf{r}_3^\uparrow/M & m_4\mathbf{r}_4^\downarrow/M & m_5\mathbf{r}_5^\uparrow/M & m_6\mathbf{r}_6^\downarrow/M \end{pmatrix}. \tag{A.2}$$

568      The relative Hamiltonian $\mathcal{H}_{\text{rel}}$ may be recast in terms of the relative Jacobi co-ordinates $\mathbf{x}$
569  $= (\mathbf{x}_1, \mathbf{x}_2, \ldots, \mathbf{x}_{N-1})^T$ (in the remainder of this section *only*, the supervector $\mathbf{x}$ excludes the
570  centre-of-mass position) [68]. The relative kinetic energy operator $T$ can be rewritten as

$$T = \sum_{i=1}^{N-1} -\frac{\hbar^2}{2\mu_i}\nabla_{\mathbf{x}_i}^2, \quad \mu_i = \left[\sum_{j=1}^{N}\frac{(\mathbb{U}_{ij})^2}{m_j}\right]^{-1}, \tag{A.3}$$

571  where $\mu_i$ is the mass associated with the Jacobi co-ordinate $\mathbf{x}_i$. Similarly, the harmonic trapping
572  potential term becomes

$$\sum_{i=1}^{N-1}\frac{\mu_i\omega_r^2}{2}\mathbf{x}_i^2, \tag{A.4}$$

573  while the two-body interaction term is transformed by reformulating the interparticle distance
574  vector:

$$\sum_{i=1}^{N}\sum_{j=i+1}^{N} V_{\text{int}}(r_{ij}), \quad r_{ij} \equiv |\mathbf{r}_i - \mathbf{r}_j| = \left[\boldsymbol{\omega}^{(ij)}\right]^T\mathbf{x}. \tag{A.5}$$

575  Above, $\boldsymbol{\omega}$ is a transformation tensor whose $(i, j)$-th component is an $(N-1)$-dimensional
576  vector with the $p$-th element given by $\left[\boldsymbol{\omega}^{(ij)}\right]_p = (\mathbb{U}^{-1})_{ip} - (\mathbb{U}^{-1})_{jp}$ [68].

577      We expand the eigenstates of the relative Hamiltonian in terms of explicitly correlated
578  Gaussian basis functions. The unsymmetrised basis functions for states with zero total relative
579  orbital angular momentum may be written as follows [68] using single-particle co-ordinates,

$$\phi_\alpha(\mathbf{y}) = \prod_{j>i=1}^{N}\exp\left[-\frac{1}{2\alpha_{ij}^2}(\mathbf{r}_i - \mathbf{r}_j)^2\right] = \exp\left[-\sum_{j>i=1}^{N}\frac{1}{2\alpha_{ij}^2}(\mathbf{r}_i - \mathbf{r}_j)^2\right], \tag{A.6}$$

580  and using Jacobi co-ordinates,

$$\phi_{\mathbb{A}}(\mathbf{x}) = \exp\left(-\frac{1}{2}\mathbf{x}^T\mathbb{A}\mathbf{x}\right), \quad \mathbb{A}_{pq} = \sum_{i=1}^{N}\sum_{j=i+1}^{N}\frac{1}{\alpha_{ij}^2}\left[\boldsymbol{\omega}^{(ij)}\right]_p\left[\boldsymbol{\omega}^{(ij)}\right]_q. \tag{A.7}$$

581  The $N(N-1)/2$ Gaussian widths $\alpha_{ij}$ are treated as non-linear variational parameters which
582  are selected semi-stochastically and optimised by minimising the energy of the state of inter-
583  est. Physically, small $\alpha_{ij}$ are required to describe contributions that occur at small interparticle
584  separations $r_{ij}$, while large $\alpha_{ij}$ are needed to describe contributions occurring at large $r_{ij}$. Due
585  to the principle of Pauli exclusion, interparticle distances are generally much longer when the
586  atom indices $i$ and $j$ correspond to identical fermions, rather than to distinguishable fermions.
587  Consequently, the $\alpha_{ij}$ parameters are generated randomly with one concession: those corre-
588  sponding to same-spin fermions are restricted to be on the order of the radial harmonic trap
589  length $l_r$, while those corresponding to different-spin fermions are permitted to range from a
590  fraction of the interaction potential width $r_0$ up to a couple of times $l_r$ [65]. Numerically, each
591  basis function is encoded as a unique $(N-1) \times (N-1)$ *correlation* matrix $\mathbb{A}$, which has the
592  properties of being real, symmetric and positive definite by virtue of the fact that the Gaussian
593  widths are positive real numbers. The property of positive definiteness ensures that the basis
594  functions are normalisable [68].

595      The correlated Gaussian technique relies on a generalisation of the variational principle
596  which accounts for excited states [38]. The basic principle states that the expectation value

of a Hamiltonian, say $\mathcal{H}_{\text{rel}}$, with respect to any normalised wave function provides an upper bound on the exact ground-state energy. If we now assume that $\varepsilon_1 \leq \varepsilon_2 \leq \cdots$ are the exact eigenenergies of $\mathcal{H}_{\text{rel}}$, and $\mathcal{E}_1 \leq \mathcal{E}_2 \leq \cdots \leq \mathcal{E}_{N_b}$ are the variational eigenvalues of $\mathcal{H}_{\text{rel}}$ obtained from the subspace spanned by $N_b$ basis functions — then the generalised principle informs us that $\varepsilon_1 \leq \mathcal{E}_1, \varepsilon_2 \leq \mathcal{E}_2, \ldots, \varepsilon_{N_b} \leq \mathcal{E}_{N_b}$. This is proven in Sec. 3.1 of Ref. [68].

For the $n^{th}$ eigenstate of $\mathcal{H}_{\text{rel}}$, the expansion in the correlated Gaussian basis (ignoring symmetrisation for now) reads,

$$\Psi_{\text{rel}}^{(n)} = \sum_{i=1}^{N_b} c_i^{(n)} \phi_{\mathbb{A}_i}(\mathbf{x}), \tag{A.8}$$

where the expansion coefficients $c_i^{(n)}$ are linear variational parameters. Minimising the variational energy $\mathcal{E}_n$ with respect to these coefficients leads to a generalised eigenvalue problem [38, 68]: $\mathbb{H}_{\text{rel}}\mathbb{C} = \mathbb{E}\mathbb{O}\mathbb{C}$. Here, $\mathbb{H}_{\text{rel}}$ and $\mathbb{O}$ are the Hamiltonian and overlap matrices, respectively, with elements given by (in two dimensions)

$$(\mathbb{H}_{\text{rel}})_{\mathbb{A}_i\mathbb{A}_j} \equiv \langle \phi_{\mathbb{A}_i} | \mathcal{H}_{\text{rel}} | \phi_{\mathbb{A}_j} \rangle, \quad \mathbb{O}_{\mathbb{A}_i\mathbb{A}_j} \equiv \langle \phi_{\mathbb{A}_i} | \phi_{\mathbb{A}_j} \rangle = \frac{(2\pi)^{N-1}}{\det[\mathbb{A}_i + \mathbb{A}_j]} \quad (i, j = 1, \ldots, N_b). \tag{A.9}$$

The $n^{th}$ lowest variational eigenvalue $\mathcal{E}_n$ corresponds to the $n^{th}$ diagonal element of the diagonal matrix $\mathbb{E}$, while the associated eigenvector $\mathbf{c}^{(n)}$ is contained in the $n^{th}$ column of the matrix $\mathbb{C}$ (not to be mistaken for the other $\mathbb{C}$ matrix defined in Appendices D and E). The generalised variational principle guarantees that $\mathcal{E}_n$ provides an upper bound on the $n^{th}$ exact eigenenergy $\varepsilon_n$ of $\mathcal{H}_{\text{rel}}$ [38, 68].

Conveniently, evaluating the matrix elements of $\mathcal{H}_{\text{rel}}$ amounts to performing simple matrix operations on $\mathbb{A}$ [68]. In two dimensions the (unsymmetrised) matrix element for the relative kinetic energy operator reads,

$$\langle \phi_{\mathbb{A}_i} | T | \phi_{\mathbb{A}_j} \rangle = \hbar^2 \text{Tr}[\mathbb{A}_i(\mathbb{A}_i + \mathbb{A}_j)^{-1}\mathbb{A}_j\mathbb{M}], \quad \mathbb{M}_{kl} = \sum_{i=1}^{N} \frac{\mathbb{U}_{ki}\mathbb{U}_{li}}{m_i} \quad (k, l = 1, \ldots, N-1). \tag{A.10}$$

The matrix elements for arbitrary one- and two-body operators are respectively given by

$$\langle \phi_{\mathbb{A}_i} | V(\mathbf{r}_k) | \phi_{\mathbb{A}_j} \rangle = \mathbb{O}_{\mathbb{A}_i\mathbb{A}_j} \frac{b_k}{2\pi} \int d^2\mathbf{r}\, V(\mathbf{r}) \exp\left(-\frac{1}{2}b_k r^2\right), \tag{A.11a}$$

$$\langle \phi_{\mathbb{A}_i} | V(\mathbf{r}_k - \mathbf{r}_l) | \phi_{\mathbb{A}_j} \rangle = \mathbb{O}_{\mathbb{A}_i\mathbb{A}_j} \frac{b_{kl}}{2\pi} \int d^2\mathbf{r}\, V(\mathbf{r}) \exp\left(-\frac{1}{2}b_{kl} r^2\right), \tag{A.11b}$$

with

$$\frac{1}{b_k} = \left[\boldsymbol{\omega}^{(k)}\right]^T (\mathbb{A}_i + \mathbb{A}_j)^{-1}\boldsymbol{\omega}^{(k)}, \quad \left[\boldsymbol{\omega}^{(k)}\right]_p = (\mathbb{U}^{-1})_{kp}, \tag{A.12a}$$

$$\frac{1}{b_{kl}} = \left[\boldsymbol{\omega}^{(kl)}\right]^T (\mathbb{A}_i + \mathbb{A}_j)^{-1}\boldsymbol{\omega}^{(kl)}, \tag{A.12b}$$

which can be used to treat the confining and interaction potentials [68]. Note that in order to endow the wave function with fermionic exchange symmetry, the antisymmetrisation operator must be acted on the unsymmetrised basis states when calculating the Hamiltonian and overlap matrix elements — and this is described in Appendix D.

We follow the two-step procedure detailed in Refs. [38, 41] to construct the explicitly correlated Gaussian basis. The first step is the basis set *enlargement*, in which new basis functions

(new matrices $\mathbb{A}_i$) are added one at a time. The second step is the basis function *refinement*, in which the existing $\mathbb{A}_i$ matrices are adjusted one at a time. Both steps are cyclically repeated as necessary until the energy of the state of interest is converged (changes by less than a preset, very small value). Due to the fact that the basis is over-complete, the rate of convergence is rapid [36].

To add a new basis function $\mathbb{A}_i$ to the basis set, we generate a large number (say '$p$') of trial basis functions stochastically within preset parameter windows: $\{\mathbb{A}_{i,1}, \mathbb{A}_{i,2}, \ldots, \mathbb{A}_{i,p}\}$. Since one more basis state always lowers the energy,[12] we choose to keep the matrix $\mathbb{A}_{i,j} \equiv \mathbb{A}_i$ that lowers the energy of the state of interest the most. Similarly, to refine an existing basis function $\mathbb{A}_i$, we generate '$p$' trial replacement basis functions stochastically: $\{\mathbb{A}'_{i,1}, \mathbb{A}'_{i,2}, \ldots, \mathbb{A}'_{i,p}\}$. We subsequently determine which one affords the lowest energy for the state of interest, labelling it by $\mathbb{A}'_{i,j} \equiv \mathbb{A}'_i$, and if this energy is lower than the original energy, then we replace $\mathbb{A}_i$ by $\mathbb{A}'_i$.

In both the enlargement and refinement phases, in order to determine how the energy eigenvalues are affected by the inclusion of a given trial basis function, we do not need to solve the full $(K+1)\times(K+1)$-dimensional generalised eigenvalue problem through matrix diagonalisation. Instead, we can exclude the concerned ($i^{th}$) row and column from the Hamiltonian and overlap matrices, and diagonalise the resulting generalised eigenvalue problem of size $K \times K$. The eigenvalues of the $(K+1)$-dimensional matrix can then be found as the roots of a secular equation which depends on the eigenvalues and normalised eigenvectors of the $K$-dimensional submatrix, and on the $i^{th}$ row and column of $\mathbb{H}_{\text{rel}}$ and $\mathbb{O}$. The full details — which are based on Gram–Schmidt orthogonalisation[13] — are provided in Ref. [32]. Selecting from a large number of trial basis functions thus becomes numerically feasible since root-finding is computationally much faster than matrix diagonalisation, and because the $K$-dimensional submatrix need only be diagonalised once. In addition, both the enlargement and refinement subroutines can be efficiently parallelised over a number ($N_c$) of MPI cores on a high-performance computer [38]. We generate $p/N_c$ trial basis functions on each core, and then compare the eigenenergies across all $N_c$ cores by using the 'MPI_Allreduce' function. Once the basis function that lowers the energy the most has been chosen, this information is synchronised across all cores by using the 'MPI_Bcast' function.

The results for $1+1$, $2+2$, and $3+3$ fermions are shown in Sec. 3. The main hindrance to theoretically considering higher particle numbers derives from the first-quantised formulation of the ECG approach — namely, the antisymmetrisation requirement to sum over all possible permutations of identical particles, as mentioned above and in Appendix D. For equally populated two-component systems of $N$ fermions, this number of permutations is $N_p = [(N/2)!]^2$, such that the evaluation of a single matrix element becomes *very* time consuming as the number of particles increases (refer to Table 1). Combined with basis sizes on the order of at least thousands of states, this makes the $6+6$ system of fermions considered by experiment [24] computationally out of reach.

| $N$ | 2 | 4 | 6 | 8 | 10 | 12 |
|---|---|---|---|---|---|---|
| $N_p$ | 1 | 4 | 36 | 576 | 14,400 | 518,400 |

Table 1: Scaling of the number of permutations $N_p$ with the number of particles $N$.

---

[12] If a basis of size $K$ yields an ordered set of eigenvalues $\lambda_1 \leq \lambda_2 \leq \cdots \leq \lambda_K$, then a basis of size $K+1$ will yield an ordered set of eigenvalues $\gamma_1 \leq \gamma_2 \leq \cdots \leq \gamma_{K+1}$, such that $\gamma_1 \leq \lambda_1 \leq \gamma_2 \leq \lambda_2 \leq \cdots \leq \gamma_K \leq \lambda_K \leq \gamma_{K+1}$.

[13] This orthogonalisation method avoids numerical instabilities caused by linear dependencies, which may otherwise arise due to the over-completeness of the basis set.

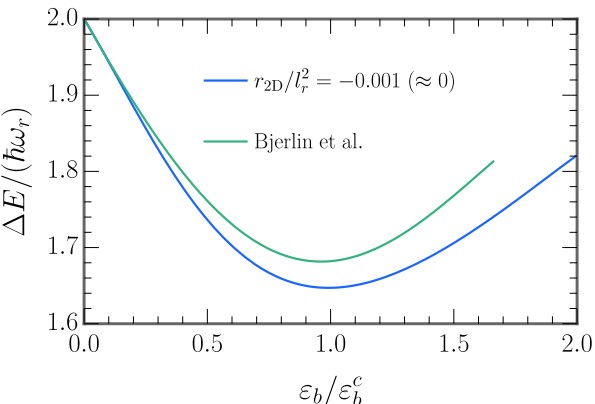

Figure 7: The lowest monopole excitation spectrum for $N_\uparrow + N_\downarrow = 3 + 3$ fermions. We overlay our result at zero effective range (in blue) on the contact interaction result from Fig. 1 of Ref. [26] (in green). In each case, we normalise $\varepsilon_b$ by a critical value $\varepsilon_b^c$, which is defined as the two-body binding energy that gives the minimum excitation energy $\Delta E$.

## B  Comparison to a Contact Interaction

The spatial extent of the potential selected to model short-range binary collisions in the ultracold Fermi gas can, to a small degree, quantitatively affect the lowest monopole excitation spectrum. Above in Fig. 7, we show again our result for $3 + 3$ fermions at an effective range of $r_{2D}/l_r^2 = -0.001 \approx 0$, which we obtained by using the finite-range Gaussian interaction potential given in Eq. (3). Although the effective range of this potential is fixed and close to zero, the physical width $r_0$ varies between $0.01 l_r$ and $0.05 l_r$ over the depicted range of binding energies. This leads to a small downward shift in the excitation energy — which becomes larger with increasing binding energy — when compared to an analogous calculation [26] based on a contact interaction with zero range ($r_0 \to 0$) [69, 70]. Within our model, we can estimate the zero-range limit of a contact interaction by starting with the value of $\Delta E$ at a particular binding energy $\varepsilon_b$, and then systematically reducing the Gaussian width $r_0$, while varying the depth $V_0$ such that $\varepsilon_b$ remains constant. In this way, we can construct a plot of $\Delta E$ versus $r_0$ and then extrapolate to the limit of $r_0 = 0$ [41]. The process can subsequently be repeated at all desired binding energies. Interestingly, due to the second term in Eq. (3), decreasing the potential width for a fixed binding energy and basis size causes $\Delta E$ to *increase*. However, since this also corresponds to a deeper potential, the result becomes less accurate. Increasing the basis size to improve the level of accuracy, in turn, lowers $\Delta E$. In general, we found that the very deep and narrow potentials generated by this limiting procedure made it necessary to use very large basis sets in order to numerically converge the excitation energy. Therefore, we only performed this check at a single binding energy.

## C  Definitions of the Pair Correlator and Density Matrices

As done in Eq. (7), we define the second-order pair correlation function for opposing spins as follows:

$$\mathcal{C}^{(2)}(\mathbf{p}_1, \mathbf{p}_2) = \langle n_\uparrow(\mathbf{p}_1) n_\downarrow(\mathbf{p}_2) \rangle - \langle n_\uparrow(\mathbf{p}_1) \rangle \langle n_\downarrow(\mathbf{p}_2) \rangle, \tag{C.1}$$

686 where

$$\widetilde{\rho}(\mathbf{p}_1, \mathbf{p}_2) \equiv \langle n_\uparrow(\mathbf{p}_1) n_\downarrow(\mathbf{p}_2) \rangle = \langle c^\dagger_{\mathbf{p}_1\uparrow} c_{\mathbf{p}_1\uparrow} c^\dagger_{\mathbf{p}_2\downarrow} c_{\mathbf{p}_2\downarrow} \rangle, \tag{C.2}$$

$$\widetilde{\rho}_\uparrow(\mathbf{p}_1) \equiv \langle n_\uparrow(\mathbf{p}_1) \rangle = \langle c^\dagger_{\mathbf{p}_1\uparrow} c_{\mathbf{p}_1\uparrow} \rangle, \tag{C.3}$$

$$\widetilde{\rho}_\downarrow(\mathbf{p}_2) \equiv \langle n_\downarrow(\mathbf{p}_2) \rangle = \langle c^\dagger_{\mathbf{p}_2\downarrow} c_{\mathbf{p}_2\downarrow} \rangle. \tag{C.4}$$

687 Here, $c^\dagger_{\mathbf{p}\sigma}$ ($c_{\mathbf{p}\sigma}$) is the fermionic creation (annihilation) operator for a particle with momentum
688 $\mathbf{p}$ and pseudospin $\sigma$ in the language of second quantisation (with $\sigma = \uparrow, \downarrow$). The "$\widetilde{\rho}$" de-
689 note momentum-space density matrix elements and these can be related to the position-space
690 density matrix elements which we have calculated in the correlated Gaussian basis (refer to
691 Appendices D and E, below).
692    To this end, we make use of the relationship between the creation operators in position
693 $[\psi^\dagger_\sigma(\mathbf{r})]$ and momentum ($c^\dagger_{\mathbf{p}\sigma}$) space:

$$c^\dagger_{\mathbf{p}\sigma} = \frac{1}{2\pi} \int d\mathbf{r}\, \psi^\dagger_\sigma(\mathbf{r})\, e^{i\mathbf{p}\cdot\mathbf{r}}, \tag{C.5}$$

$$c_{\mathbf{p}\sigma} = \frac{1}{2\pi} \int d\mathbf{r}\, \psi_\sigma(\mathbf{r})\, e^{-i\mathbf{p}\cdot\mathbf{r}}. \tag{C.6}$$

694 Inserting these relations into the definition (C.3) of the one-body density matrix for the spin-$\uparrow$
695 atoms in momentum space yields

$$\widetilde{\rho}_\uparrow(\mathbf{p}_1) = \frac{1}{(2\pi)^2} \int\int d\mathbf{r}\, d\mathbf{r}'\, \langle \psi^\dagger_\uparrow(\mathbf{r}) \psi_\uparrow(\mathbf{r}') \rangle\, e^{-i\mathbf{p}_1\cdot(\mathbf{r}'-\mathbf{r})} = \frac{1}{(2\pi)^2} \int\int d\mathbf{r}\, d\mathbf{r}'\, \rho_\uparrow(\mathbf{r}, \mathbf{r}')\, e^{-i\mathbf{p}_1\cdot(\mathbf{r}'-\mathbf{r})}. \tag{C.7}$$

696 This result involves the position-space one-body density matrix for the spin-$\uparrow$ atoms, which
697 can be written as

$$\rho_\uparrow(\mathbf{r}, \mathbf{r}') = \left[ \int\cdots\int d\mathbf{r}^\uparrow_1 d\mathbf{r}^\downarrow_2 \cdots d\mathbf{r}^\uparrow_{N-1} d\mathbf{r}^\downarrow_N \left| \Psi(\mathbf{r}^\uparrow_1, \mathbf{r}^\downarrow_2, \cdots, \mathbf{r}^\uparrow_{N-1}, \mathbf{r}^\downarrow_N) \right|^2 \right]^{-1} \times$$

$$\int\cdots\int d\mathbf{r}^\downarrow_2 d\mathbf{r}^\uparrow_3 d\mathbf{r}^\downarrow_4 \cdots d\mathbf{r}^\uparrow_{N-1} d\mathbf{r}^\downarrow_N \Psi(\mathbf{r}, \mathbf{r}^\downarrow_2, \mathbf{r}^\uparrow_3, \mathbf{r}^\downarrow_4, \cdots, \mathbf{r}^\uparrow_{N-1}, \mathbf{r}^\downarrow_N) \Psi^*(\mathbf{r}', \mathbf{r}^\downarrow_2, \mathbf{r}^\uparrow_3, \mathbf{r}^\downarrow_4, \cdots, \mathbf{r}^\uparrow_{N-1}, \mathbf{r}^\downarrow_N) \tag{C.8}$$

698 in the first quantisation picture, where $\Psi = \Psi_{\text{com}} \Psi_{\text{rel}}$ is the total $N$-body wave function. The
699 first line of Eq. (C.8) is a normalisation constant; in the second line we integrate the density
700 $\Psi\Psi^*$ over all co-ordinates except those of a single spin-$\uparrow$ particle. Expressions analogous to
701 Eqs. (C.7)–(C.8) can readily be written down for the spin-$\downarrow$ case (C.4). Similarly, the two-body
702 density matrix for spin-$\uparrow$-spin-$\downarrow$ pairs is given by

$$\widetilde{\rho}(\mathbf{p}_1, \mathbf{p}_2) = \frac{1}{(2\pi)^4} \int\cdots\int d\mathbf{r}_1 d\mathbf{r}'_1 d\mathbf{r}_2 d\mathbf{r}'_2 \langle \psi^\dagger_\uparrow(\mathbf{r}_1) \psi_\uparrow(\mathbf{r}'_1) \psi^\dagger_\downarrow(\mathbf{r}_2) \psi_\downarrow(\mathbf{r}'_2) \rangle\, e^{-i\mathbf{p}_1\cdot(\mathbf{r}'_1-\mathbf{r}_1)} e^{-i\mathbf{p}_2\cdot(\mathbf{r}'_2-\mathbf{r}_2)}$$

$$= \frac{1}{(2\pi)^4} \int\cdots\int d\mathbf{r}_1 d\mathbf{r}'_1 d\mathbf{r}_2 d\mathbf{r}'_2\, \rho(\mathbf{r}_1, \mathbf{r}'_1; \mathbf{r}_2, \mathbf{r}'_2)\, e^{-i\mathbf{p}_1\cdot(\mathbf{r}'_1-\mathbf{r}_1)} e^{-i\mathbf{p}_2\cdot(\mathbf{r}'_2-\mathbf{r}_2)} \tag{C.9}$$

703 in momentum space, and by

$$\rho(\mathbf{r}_1, \mathbf{r}'_1; \mathbf{r}_2, \mathbf{r}'_2) = \left[ \int\cdots\int d\mathbf{r}^\uparrow_1 d\mathbf{r}^\downarrow_2 \cdots d\mathbf{r}^\uparrow_{N-1} d\mathbf{r}^\downarrow_N \left| \Psi(\mathbf{r}^\uparrow_1, \mathbf{r}^\downarrow_2, \cdots, \mathbf{r}^\uparrow_{N-1}, \mathbf{r}^\downarrow_N) \right|^2 \right]^{-1} \times$$

$$\int \cdots \int d\mathbf{r}_3^\uparrow d\mathbf{r}_4^\downarrow \cdots d\mathbf{r}_{N-1}^\uparrow d\mathbf{r}_N^\downarrow \Psi(\mathbf{r}_1, \mathbf{r}_2, \mathbf{r}_3^\uparrow, \mathbf{r}_4^\downarrow, \cdots, \mathbf{r}_{N-1}^\uparrow, \mathbf{r}_N^\downarrow) \Psi^*(\mathbf{r}_1', \mathbf{r}_2', \mathbf{r}_3^\uparrow, \mathbf{r}_4^\downarrow, \cdots, \mathbf{r}_{N-1}^\uparrow, \mathbf{r}_N^\downarrow)$$

(C.10)

in position space. Above, we integrate over all co-ordinates except those of one spin-$\uparrow$ particle and one spin-$\downarrow$ particle. Note that all integrals in this section are two-dimensional, i.e, we have written $d\mathbf{r} \equiv d^2\mathbf{r}$ for brevity. Furthermore, for numerical convenience we order the atoms so that the first one is spin-$\uparrow$, the second is spin-$\downarrow$, the third is spin-$\uparrow$, etc., as done in Appendix A [see Eqs. (A.1)–(A.2)].

## D  Derivation of the One-Body Terms in the Pair Correlator

To derive closed analytical expressions for the one-body terms in Eq. (C.1), we follow the prescription given in Appendix A of Ref. [39] (which is in three dimensions), while making the necessary modifications for a two-dimensional system.

When we calculated the excitation spectra in Fig. 2, we separated off the centre-of-mass degrees of freedom and expanded the eigenstates of the relative Hamiltonian in terms of the explicitly correlated Gaussian basis functions. These basis functions depended on a set of non-linear variational parameters which were optimised through energy minimisation. In order to calculate the pair correlator $C^{(2)}$ we now need to utilise the full $N$-body wave function, so we multiply the optimised basis set by the unnormalised ground-state centre-of-mass wave function [39]:

$$\Psi_{\mathrm{com}}^{\mathrm{(GS)}}(\mathbf{x}_N) = \exp\left(-\frac{\mathbf{x}_N^2}{2a_{\mathrm{ho}}^2/N}\right), \quad \mathbf{x}_N = \sum_{i=1}^N \frac{\mathbf{r}_i^\sigma}{N}.$$

(D.1)

The unsymmetrised (and unnormalised) basis functions that incorporate the centre-of-mass motion thus read as follows:

$$\phi_{\mathbb{A}}(\mathbf{x}) = \exp\left(-\frac{1}{2}\mathbf{x}^T \mathbb{A} \mathbf{x}\right),$$

(D.2)

where $\mathbf{x} = (\mathbf{x}_1, \mathbf{x}_2, \ldots, \mathbf{x}_{N-1}, \mathbf{x}_N)$ denotes the *full* set of $N$ Jacobi position vectors defined in Appendix A. Here, $\mathbb{A}$ is an $N \times N$ symmetric and positive definite correlation matrix comprising $N(N-1)/2$ variational parameters (the $\mathbb{A}_{ij}$ with $i = 1, \ldots, N-1$ and $j \geq i$), which are optimised semi-stochastically. To force the centre-of-mass degrees of freedom into the ground state, we manually set the matrix elements $\mathbb{A}_{iN}$ and $\mathbb{A}_{Ni}$ (with $i = 1, \ldots, N-1$) to zero, while setting $\mathbb{A}_{NN}$ to $N/a_{\mathrm{ho}}^2$ [39]. We reiterate that $\mathbf{x}$ is a 'supervector' (or vector of vectors) and the double-line font is used in this work to designate a matrix. The Jacobi vectors $\mathbf{x}$ and single-particle co-ordinates $\mathbf{y} \equiv (\mathbf{y}_1, \ldots, \mathbf{y}_N) = (\mathbf{r}_1^\uparrow, \mathbf{r}_2^\downarrow, \mathbf{r}_3^\uparrow, \ldots, \mathbf{r}_N^\downarrow)$ are related by the $N \times N$ linear transformation matrix $\mathbb{U}$, which has been defined in Eqs. (A.1)–(A.2) of Appendix A.

Now that we have set up the system, our first goal is to derive the correlated Gaussian matrix elements of the real-space one-body density matrix for the spin-$\uparrow$ atoms, Eq. (C.8) (the derivation for the spin-$\downarrow$ atoms follows analogously):

$$\frac{[\rho_\uparrow(\mathbf{r}, \mathbf{r}')]_{\mathbb{A}\mathbb{A}'}}{\mathbb{O}_{\mathbb{A}\mathbb{A}'}} \equiv \frac{\langle \phi_{\mathbb{A}} | \rho_\uparrow | \phi_{\mathbb{A}'} \rangle}{\langle \phi_{\mathbb{A}} | \phi_{\mathbb{A}'} \rangle}$$

$$= (\mathbb{O}_{\mathbb{A}\mathbb{A}'})^{-1} \int \cdots \int d^{2N-2}\mathbf{y}_{\mathrm{red}} \left[\int d^2\mathbf{r}_1^\uparrow \delta(\mathbf{r} - \mathbf{r}_1^\uparrow) \phi_{\mathbb{A}}(\mathbf{x})\right]\left[\int d^2\mathbf{r}_1^\uparrow \delta(\mathbf{r}' - \mathbf{r}_1^\uparrow) \phi_{\mathbb{A}'}(\mathbf{x})\right]. \quad \text{(D.3)}$$

In this equation we have defined $\mathbf{y}_{\text{red}} = (\mathbf{r}_2^\downarrow, \mathbf{r}_3^\uparrow, \mathbf{r}_4^\downarrow, \ldots, \mathbf{r}_{N-1}^\uparrow, \mathbf{r}_N^\downarrow)$, $\delta(\cdots)$ represents the two-dimensional Dirac delta function, and

$$\mathbb{O}_{\mathbb{A}\mathbb{A}'} \equiv \langle \phi_\mathbb{A} | \phi_{\mathbb{A}'} \rangle = \frac{(2\pi)^N}{\det[\mathbb{A} + \mathbb{A}']} \tag{D.4}$$

is the overlap matrix element [68] for the (unsymmetrised) ECG basis functions associated with the correlation matrices $\mathbb{A}$ and $\mathbb{A}'$. It is convenient to express the right-hand-side of Eq. (D.3) in terms of the Gaussian generating function [68],

$$g(\mathbf{s}; \mathbb{A}, \mathbf{x}) = \exp\left(-\frac{1}{2}\mathbf{x}^T \mathbb{A} \mathbf{x} + \mathbf{s}^T \mathbf{x}\right), \tag{D.5}$$

where $\mathbf{s}$ denotes an auxiliary supervector with the same dimensionality as $\mathbf{x}$. The basis function in Eq. (D.2) can therefore be written as $\phi_\mathbb{A}(\mathbf{x}) = g(\mathbf{0}; \mathbb{A}, \mathbf{x})$. By using the fact that $\mathbf{x}^T \mathbb{A} \mathbf{x} = \mathbf{y}^T \mathbb{U}^T \mathbb{A} \mathbb{U} \mathbf{y}$, we re-express the basis function $\phi_\mathbb{A}$ in terms of $\mathbf{y}$ and separate off the $\mathbf{r}_1^\uparrow$ dependence:

$$\phi_\mathbb{A}(\mathbf{y}) = g(\mathbf{0}; \mathbb{B}, \mathbf{y}_{\text{red}}) \exp\left[-\frac{1}{2}b_1(\mathbf{r}_1^\uparrow)^2 - (\mathbf{b}^T \mathbf{y}_{\text{red}})^T \mathbf{r}_1^\uparrow\right]. \tag{D.6}$$

Here, $\mathbb{B}$ is an $(N-1)\times(N-1)$-dimensional matrix given by $\mathbb{U}^T \mathbb{A} \mathbb{U}$ with the first row and column removed, $\mathbf{b}$ is an $(N-1)$-dimensional vector given by $((\mathbb{U}^T \mathbb{A} \mathbb{U})_{12}, \ldots, (\mathbb{U}^T \mathbb{A} \mathbb{U})_{1N})$, and $b_1$ is a scalar given by $(\mathbb{U}^T \mathbb{A} \mathbb{U})_{11}$. In addition, Eq. (D.6) contains the quantity

$$(\mathbf{b}^T \mathbf{y}_{\text{red}})^T \mathbf{r}_1^\uparrow = \sum_{j=2}^N \mathbf{b}_{j-1} \mathbf{y}_j^T \mathbf{r}_1^\uparrow, \tag{D.7}$$

where $\mathbf{b}_j$ denotes the $j^{th}$ element of the vector $\mathbf{b}$. To continue we define $\{\mathbb{B}', \mathbf{b}', b_1'\}$ analogously to $\{\mathbb{B}, \mathbf{b}, b_1\}$, substitute the expressions for $\phi_\mathbb{A}(\mathbf{x}) \to \phi_\mathbb{A}(\mathbf{y})$ and $\phi_{\mathbb{A}'}(\mathbf{x}) \to \phi_{\mathbb{A}'}(\mathbf{y})$ into Eq. (D.3), and then evaluate the two Dirac delta functions. This yields

$$[\rho_\uparrow(\mathbf{r}, \mathbf{r}')]_{\mathbb{A}\mathbb{A}'} \equiv \langle \phi_\mathbb{A} | \rho_\uparrow | \phi_{\mathbb{A}'} \rangle = \int \cdots \int d^{2N-2}\mathbf{y}_{\text{red}}\, g(\mathbf{0}; \mathbb{B}, \mathbf{y}_{\text{red}})\, g(\mathbf{0}; \mathbb{B}', \mathbf{y}_{\text{red}}) \times$$
$$\exp\left\{-\frac{1}{2}b_1 \mathbf{r}^2 - \frac{1}{2}b_1'(\mathbf{r}')^2 - (\mathbf{b}^T \mathbf{y}_{\text{red}})^T \mathbf{r} - [(\mathbf{b}')^T \mathbf{y}_{\text{red}}]^T \mathbf{r}'\right\}, \tag{D.8}$$

which can be rewritten as

$$[\rho_\uparrow(\mathbf{r}, \mathbf{r}')]_{\mathbb{A}\mathbb{A}'} = \int \cdots \int d^{2N-2}\mathbf{y}_{\text{red}}\, g[-(\mathbf{br} + \mathbf{b}'\mathbf{r}'); \mathbb{B} + \mathbb{B}', \mathbf{y}_{\text{red}}] \exp\left\{-\frac{1}{2}\left[b_1 \mathbf{r}^2 + b_1'(\mathbf{r}')^2\right]\right\}. \tag{D.9}$$

Above, the quantity $\mathbf{br}$ is an $(N-1)$-dimensional supervector with elements $\mathbf{b}_j \mathbf{r}$, where $j = 1, \ldots, N-1$. By employing the two-dimensional relation [68] shown below,

$$\int \cdots \int d^{2N}\mathbf{x}\, g(\mathbf{s}; \mathbb{A}, \mathbf{x}) = \frac{(2\pi)^N}{\det[\mathbb{A}]} \exp\left(\frac{1}{2}\mathbf{s}^T \mathbb{A}^{-1} \mathbf{s}\right), \tag{D.10}$$

we arrive at a compact expression for the correlated Gaussian matrix elements of the one-body density matrix in real space:

$$[\rho_\uparrow(\mathbf{r}, \mathbf{r}')]_{\mathbb{A}\mathbb{A}'} = c_1 \exp\left\{-\frac{1}{2}\left[c\mathbf{r}^2 + c'(\mathbf{r}')^2 - a\mathbf{r}^T \mathbf{r}'\right]\right\}, \tag{D.11}$$

which depends on the following scalars,

$$c_1 = \frac{(2\pi)^{N-1}}{\det[\mathbb{B} + \mathbb{B}']}, \tag{D.12}$$

$$c = b_1 - \mathbf{b}^T \mathbb{C} \mathbf{b}, \tag{D.13}$$

$$c' = b_1' - (\mathbf{b}')^T \mathbb{C} \mathbf{b}', \tag{D.14}$$

$$a = \mathbf{b}^T \mathbb{C} \mathbf{b}' + (\mathbf{b}')^T \mathbb{C} \mathbf{b}, \tag{D.15}$$

and on the matrix,

$$\mathbb{C} = (\mathbb{B} + \mathbb{B}')^{-1}. \tag{D.16}$$

Our second goal is now to evaluate the Fourier transform of Eq. (D.11) — as defined by Eq. (C.7) — in order to obtain the correlated Gaussian matrix elements of the one-body density matrix in momentum space:

$$[\widetilde{\rho}_\uparrow(\mathbf{p}_1)]_{\mathbb{A}\mathbb{A}'} = \frac{1}{(2\pi)^2} \int \int d^2\mathbf{r}\, d^2\mathbf{r}'\, [\rho_\uparrow(\mathbf{r}, \mathbf{r}')]_{\mathbb{A}\mathbb{A}'}\, e^{-i\mathbf{p}_1 \cdot (\mathbf{r}' - \mathbf{r})}. \tag{D.17}$$

By defining $\mathbf{X} = \mathbf{r}' - \mathbf{r}$, Eq. (D.17) becomes

$$[\widetilde{\rho}_\uparrow(\mathbf{p}_1)]_{\mathbb{A}\mathbb{A}'} = \frac{c_1}{(2\pi)^2} \int \int d^2\mathbf{r}\, d^2\mathbf{X}\, \exp[-i(p_1^x X_x + p_1^y X_y)] \times$$
$$\exp\left\{ \frac{1}{2}\left[ g_1(r_x^2 + r_y^2) + g_2(X_x^2 + X_y^2) + g_3(r_x X_x + r_y X_y) \right] \right\}, \tag{D.18}$$

with the scalars,

$$g_1 = a - c - c', \tag{D.19}$$

$$g_2 = -c', \tag{D.20}$$

$$g_3 = a - 2c'. \tag{D.21}$$

For $g_1 < 0$, the integral over $\mathbf{r}$ can be performed analytically:

$$\int_{-\infty}^{+\infty} \int_{-\infty}^{+\infty} dr_x\, dr_y\, \exp\left\{ \frac{1}{2}\left[ g_1(r_x^2 + r_y^2) + g_2(X_x^2 + X_y^2) + g_3(r_x X_x + r_y X_y) \right] \right\}$$
$$= -\frac{2\pi}{g_1} \exp\left[ \frac{4g_1 g_2 - g_3^2}{8g_1} (X_x^2 + X_y^2) \right]. \tag{D.22}$$

This allows the integral over $\mathbf{X}$ to then be carried out analytically, as well, for $4g_1 g_2 - g_3^2 > 0$:

$$\int_{-\infty}^{+\infty} \int_{-\infty}^{+\infty} dX_x\, dX_y\, \exp[-i(p_1^x X_x + p_1^y X_y)] \left\{ -\frac{2\pi}{g_1} \exp\left[ \frac{4g_1 g_2 - g_3^2}{8g_1} (X_x^2 + X_y^2) \right] \right\}$$
$$= \frac{16\pi^2}{4g_1 g_2 - g_3^2} \exp\left\{ \frac{2g_1}{4g_1 g_2 - g_3^2} \left[ (p_1^x)^2 + (p_1^y)^2 \right] \right\}. \tag{D.23}$$

Thus, the correlated Gaussian matrix elements of the momentum-space one-body density matrix for the spin-$\uparrow$ atoms are given by

$$[\widetilde{\rho}_\uparrow(\mathbf{p}_1)]_{\mathbb{A}\mathbb{A}'} = \frac{4c_1}{4g_1 g_2 - g_3^2} \exp\left( \frac{2g_1}{4g_1 g_2 - g_3^2} p_1^2 \right), \tag{D.24}$$

with momentum $p_1 \equiv |\mathbf{p}_1|$. We have checked that the two conditions, $g_1 < 0$ and $4g_1g_2 - g_3^2 > 0$, are indeed satisfied numerically. We can now evaluate Eq. (C.3) for the ground state (GS) by using the derived results for $[\tilde{\rho}_\uparrow(\mathbf{p}_1)]_{\mathbb{A}_i\mathbb{A}_j}$ (D.24) and $\mathbb{O}_{\mathbb{A}_i\mathbb{A}_j}$ (D.4):

$$\langle n_\uparrow(\mathbf{p}_1)\rangle \equiv \frac{\langle \Psi^{(GS)}|n_\uparrow(\mathbf{p}_1)|\Psi^{(GS)}\rangle}{\langle \Psi^{(GS)}|\Psi^{(GS)}\rangle} = \frac{\sum_{i,j} c_i^* [\tilde{\rho}_\uparrow(\mathbf{p}_1)]_{\mathbb{A}_i\mathbb{A}_j} c_j}{\sum_{i,j} c_i^* \mathbb{O}_{\mathbb{A}_i\mathbb{A}_j} c_j}. \tag{D.25}$$

Above, the second expression is obtained from the first by inserting two complete sets of ECG basis states into both the numerator and denominator, and $c_i = \langle \phi_{\mathbb{A}_i}|\Psi^{(GS)}\rangle$ is the $i^{th}$ (real) coefficient of the full ground-state wave function which is found by diagonalising the Hamiltonian (see Appendix A).

To enhance the clarity of our discussion up until this point, we have used unsymmetrised basis functions — but of course, in reality, when we derive the ECG matrix elements we need to appropriately *antisymmetrise* the fermionic basis [68]. This means that we need to act the antisymmetrisation operator,

$$\mathcal{P} = \sum_{i=1}^{N_p} s_i P_i, \tag{D.26}$$

on both the bra $\langle \phi_{\mathbb{A}}|$ and the ket $|\phi_{\mathbb{A}'}\rangle$. Here, $\mathcal{P}$ represents the sum of all possible $N_p$ permutation operators $P_i$ for the reordering of identical fermions, weighted by the signs $s_i$ of those permutations. Conveniently, in the ECG approach acting a single permutation operator on a basis function simply amounts to a redefinition of the correlation matrix $\mathbb{A} \to \bar{\mathbb{A}}(i)$:

$$P_i \phi_{\mathbb{A}}(\mathbf{x}) = P_i \exp\left(-\frac{1}{2}\mathbf{x}^T \mathbb{A}\mathbf{x}\right) = \exp\left\{-\frac{1}{2}\mathbf{x}^T \left[\left(\mathbb{T}_{P_i}\right)^T \mathbb{A}\mathbb{T}_{P_i}\right]\mathbf{x}\right\} \equiv \exp\left[-\frac{1}{2}\mathbf{x}^T \bar{\mathbb{A}}(i)\mathbf{x}\right] = \phi_{\bar{\mathbb{A}}(i)}(\mathbf{x}), \tag{D.27}$$

where $\mathbb{T}_{P_i}$ is the $(N-1) \times (N-1)$-dimensional permutation matrix corresponding to the $i^{th}$ permutation — as defined in Eq. (2.30) of Ref. [68]. Accordingly, it is straightforward to write down the antisymmetrised matrix element of a given operator, say $\mathcal{B}$:

$$\langle \phi_{\mathbb{A}}|\mathcal{B}|\phi_{\mathbb{A}'}\rangle \to \langle \mathcal{P}\phi_{\mathbb{A}}|\mathcal{B}|\mathcal{P}\phi_{\mathbb{A}'}\rangle = \sum_{i=1}^{N_p}\sum_{j=1}^{N_p} s_i s_j \langle \phi_{\bar{\mathbb{A}}(i)}|\mathcal{B}|\phi_{\bar{\mathbb{A}}'(j)}\rangle, \tag{D.28}$$

which comprises $N_p^2$ terms. If $\mathcal{B}$ is invariant under the exchange of any pair of identical atoms (i.e., if it commutes with all permutation operators $P_i$), then we can use this fact — and also the fact that each permutation is an idempotent operator, $(P_i)^2 = 1 \ \forall \ i$ — to show that

$$\langle \mathcal{P}\phi_{\mathbb{A}}|\mathcal{B}|\mathcal{P}\phi_{\mathbb{A}'}\rangle = N_p\langle \mathcal{P}\phi_{\mathbb{A}}|\mathcal{B}|\phi_{\mathbb{A}'}\rangle = N_p\langle \phi_{\mathbb{A}}|\mathcal{B}|\mathcal{P}\phi_{\mathbb{A}'}\rangle. \tag{D.29}$$

Now, the right-hand side is a sum of only $N_p$ terms. These operator conditions are clearly satisfied by the identity, and hence, the overlap matrix element in the denominator of Eq. (D.3) can be antisymmetrised as follows:

$$\mathbb{O}_{\mathbb{A}\mathbb{A}'} \equiv \langle \phi_{\mathbb{A}}|\phi_{\mathbb{A}'}\rangle \to N_p\langle \phi_{\mathbb{A}}|\mathcal{P}\phi_{\mathbb{A}'}\rangle = N_p \sum_{j=1}^{N_p} s_j \langle \phi_{\mathbb{A}}|\phi_{\bar{\mathbb{A}}'(j)}\rangle = N_p \sum_{j=1}^{N_p} \frac{s_j (2\pi)^N}{\det[\mathbb{A} + \bar{\mathbb{A}}'(j)]}. \tag{D.30}$$

Equation (D.29) additionally holds for the Hamiltonian $\mathcal{H}$ in Eq. (1), but not for the density matrices in Appendix C, and thus the numerator of Eq. (D.3) must be antisymmetrised by using Eq. (D.28). Calculations of structural properties are consequently much longer than those of

energy and excitation spectra. Note that the redefined correlation matrices $\bar{\mathbb{A}}(i)$ and $\bar{\mathbb{A}}'(j)$ will affect the values of the $\mathbb{B}$ matrix, $\mathbf{b}$ vector, and $b_1$ scalar first appearing in Eq. (D.6) (as well as their primed equivalents), and all subsequent quantities that depend on these. Equation (D.29) is very useful since in the ECG method, the principal limiting factor on computational time for increasing particle number $N$ is the number of permutations $N_p$ required to antisymmetrise the wave function, as we discussed in Appendix A.

# E   Derivation of the Two-Body Term in the Pair Correlator

We can directly extend the approach in Appendix D to derive a closed analytical expression for the two-body term in Eq. (C.1). To this end, we consider the two-body density matrix for spin-↑-spin-↓ pairs in real space, Eq. (C.10), and we calculate its matrix elements in the explicitly correlated Gaussian basis. The two-body equivalent of Eq. (D.3) is shown below:

$$
\begin{aligned}
\frac{[\rho(\mathbf{r}_\uparrow, \mathbf{r}'_\uparrow; \mathbf{r}_\downarrow, \mathbf{r}'_\downarrow)]_{\mathbb{A}\mathbb{A}'}}{\mathbb{O}_{\mathbb{A}\mathbb{A}'}} &\equiv \frac{\langle \phi_{\mathbb{A}} | \rho | \phi_{\mathbb{A}'} \rangle}{\langle \phi_{\mathbb{A}} | \phi_{\mathbb{A}'} \rangle} \\
&= (\mathbb{O}_{\mathbb{A}\mathbb{A}'})^{-1} \int \cdots \int d^{2N-4}\mathbf{y}_{\mathrm{red}} \left[ \int \int d^2\mathbf{r}_1^\uparrow d^2\mathbf{r}_2^\downarrow \, \delta(\mathbf{r}_\uparrow - \mathbf{r}_1^\uparrow)\, \delta(\mathbf{r}_\downarrow - \mathbf{r}_2^\downarrow)\, \phi_{\mathbb{A}}(\mathbf{x}) \right] \times \\
&\qquad\qquad \left[ \int \int d^2\mathbf{r}_1^\uparrow d^2\mathbf{r}_2^\downarrow \, \delta(\mathbf{r}'_\uparrow - \mathbf{r}_1^\uparrow)\, \delta(\mathbf{r}'_\downarrow - \mathbf{r}_2^\downarrow)\, \phi_{\mathbb{A}'}(\mathbf{x}) \right],
\end{aligned}
\tag{E.1}
$$

where now $\mathbf{y}_{\mathrm{red}} = (\mathbf{r}_3^\uparrow, \mathbf{r}_4^\downarrow, \ldots, \mathbf{r}_{N-1}^\uparrow, \mathbf{r}_N^\downarrow)$, while $\mathbb{O}_{\mathbb{A}\mathbb{A}'} \equiv \langle \phi_{\mathbb{A}} | \phi_{\mathbb{A}'} \rangle$ is still defined by Eq. (D.4). By using Eqs. (A.1) and (D.5), we rewrite the basis function $\phi_{\mathbb{A}}$ in terms of $\mathbf{y}$ and separate off the $\mathbf{r}_1^\uparrow$ and $\mathbf{r}_2^\downarrow$ dependencies:

$$
\phi_{\mathbb{A}}(\mathbf{y}) = g(\mathbf{0}; \mathbb{B}, \mathbf{y}_{\mathrm{red}}) \exp\left[ -\frac{1}{2} b_1 (\mathbf{r}_1^\uparrow)^2 - \frac{1}{2} b_2 (\mathbf{r}_2^\downarrow)^2 - b_3 (\mathbf{r}_1^\uparrow)^T \mathbf{r}_2^\downarrow - (\mathbf{b}_1^T \mathbf{y}_{\mathrm{red}})^T \mathbf{r}_1^\uparrow - (\mathbf{b}_2^T \mathbf{y}_{\mathrm{red}})^T \mathbf{r}_2^\downarrow \right].
\tag{E.2}
$$

Above, the $(N-2) \times (N-2)$-dimensional matrix $\mathbb{B}$ is given by $\mathbb{U}^T \mathbb{A} \mathbb{U}$ with the first and second rows and columns removed. Equation (E.2) additionally contains two $(N-2)$-dimensional vectors:

$$
\mathbf{b}_1 = ((\mathbb{U}^T \mathbb{A} \mathbb{U})_{13}, (\mathbb{U}^T \mathbb{A} \mathbb{U})_{14}, \ldots, (\mathbb{U}^T \mathbb{A} \mathbb{U})_{1N}),
\tag{E.3}
$$

$$
\mathbf{b}_2 = ((\mathbb{U}^T \mathbb{A} \mathbb{U})_{23}, (\mathbb{U}^T \mathbb{A} \mathbb{U})_{24}, \ldots, (\mathbb{U}^T \mathbb{A} \mathbb{U})_{2N}),
\tag{E.4}
$$

and three scalars: $b_1 = (\mathbb{U}^T \mathbb{A} \mathbb{U})_{11}$, $b_2 = (\mathbb{U}^T \mathbb{A} \mathbb{U})_{22}$, $b_3 = (\mathbb{U}^T \mathbb{A} \mathbb{U})_{12}$. To be clear, we mention that

$$
(\mathbf{b}_i^T \mathbf{y}_{\mathrm{red}})^T \mathbf{r}_i^\sigma = \sum_{j=3}^N (\mathbf{b}_i)_{j-2} \mathbf{y}_j^T \mathbf{r}_i^\sigma,
\tag{E.5}
$$

where $(\mathbf{b}_i)_j$ denotes the $j^{th}$ element of the vector $\mathbf{b}_i$ (with $i = 1, 2$). We also define analogous quantities $\{\mathbb{B}', \mathbf{b}'_1, \mathbf{b}'_2, b'_1, b'_2, b'_3\}$ which correspond to the basis function $\phi_{\mathbb{A}'}$. To proceed, we substitute the expressions for $\phi_{\mathbb{A}}(\mathbf{x}) \to \phi_{\mathbb{A}}(\mathbf{y})$ and $\phi_{\mathbb{A}'}(\mathbf{x}) \to \phi_{\mathbb{A}'}(\mathbf{y})$ into Eq. (E.1), and then evaluate the four Dirac delta functions. This gives

$$
[\rho(\mathbf{r}_\uparrow, \mathbf{r}'_\uparrow; \mathbf{r}_\downarrow, \mathbf{r}'_\downarrow)]_{\mathbb{A}\mathbb{A}'} \equiv \langle \phi_{\mathbb{A}} | \rho | \phi_{\mathbb{A}'} \rangle = \int \cdots \int d^{2N-4}\mathbf{y}_{\mathrm{red}}\, g(\mathbf{0}; \mathbb{B}, \mathbf{y}_{\mathrm{red}})\, g(\mathbf{0}; \mathbb{B}', \mathbf{y}_{\mathrm{red}}) \times
$$

$$\exp\left\{-\frac{1}{2}b_1\mathbf{r}_\uparrow^2 - \frac{1}{2}b_2\mathbf{r}_\downarrow^2 - b_3\mathbf{r}_\uparrow^T\mathbf{r}_\downarrow - (\mathbf{b}_1^T\mathbf{y}_{\text{red}})^T\mathbf{r}_\uparrow - (\mathbf{b}_2^T\mathbf{y}_{\text{red}})^T\mathbf{r}_\downarrow\right\}\times$$

$$\exp\left\{-\frac{1}{2}b_1'(\mathbf{r}_\uparrow')^2 - \frac{1}{2}b_2'(\mathbf{r}_\downarrow')^2 - b_3'(\mathbf{r}_\uparrow')^T\mathbf{r}_\downarrow' - [(\mathbf{b}_1')^T\mathbf{y}_{\text{red}}]^T\mathbf{r}_\uparrow' - [(\mathbf{b}_2')^T\mathbf{y}_{\text{red}}]^T\mathbf{r}_\downarrow'\right\}, \tag{E.6}$$

815 which can be reformulated as

$$[\rho(\mathbf{r}_\uparrow, \mathbf{r}_\uparrow'; \mathbf{r}_\downarrow, \mathbf{r}_\downarrow')]_{\mathbb{A}\mathbb{A}'} = \int d^{2N-4}\mathbf{y}_{\text{red}}\, g[-(\mathbf{b}_1\mathbf{r}_\uparrow + \mathbf{b}_2\mathbf{r}_\downarrow + \mathbf{b}_1'\mathbf{r}_\uparrow' + \mathbf{b}_2'\mathbf{r}_\downarrow'); \mathbb{B} + \mathbb{B}', \mathbf{y}_{\text{red}}]\times$$

$$\exp\left\{-\left[\frac{1}{2}b_1\mathbf{r}_\uparrow^2 + \frac{1}{2}b_2\mathbf{r}_\downarrow^2 + b_3\mathbf{r}_\uparrow^T\mathbf{r}_\downarrow + \frac{1}{2}b_1'(\mathbf{r}_\uparrow')^2 + \frac{1}{2}b_2'(\mathbf{r}_\downarrow')^2 + b_3'(\mathbf{r}_\uparrow')^T\mathbf{r}_\downarrow'\right]\right\}. \tag{E.7}$$

816 Here $\mathbf{b}_1\mathbf{r}_\uparrow$, for instance, is an $(N-2)$-dimensional supervector with elements $(\mathbf{b}_1)_j\mathbf{r}_\uparrow$, where
817 $j = 1, \ldots, N-2$. By applying the identity in Eq. (D.10), we can solve the integral over $\mathbf{y}_{\text{red}}$ to
818 yield an expression for the ECG matrix elements of the two-body density matrix in real space:

$$[\rho(\mathbf{r}_\uparrow, \mathbf{r}_\uparrow'; \mathbf{r}_\downarrow, \mathbf{r}_\downarrow')]_{\mathbb{A}\mathbb{A}'} = a_1\exp\left\{-\frac{1}{2}\left[c_1\mathbf{r}_\uparrow^2 + c_1'(\mathbf{r}_\uparrow')^2 + c_2\mathbf{r}_\downarrow^2 + c_2'(\mathbf{r}_\downarrow')^2 + d_1\mathbf{r}_\uparrow^T\mathbf{r}_\downarrow + \right.\right.$$

$$\left.\left. d_1'(\mathbf{r}_\uparrow')^T\mathbf{r}_\downarrow' - f_1\mathbf{r}_\uparrow^T\mathbf{r}_\uparrow' - f_2\mathbf{r}_\downarrow^T\mathbf{r}_\downarrow' - f_3\mathbf{r}_\uparrow^T\mathbf{r}_\downarrow' - f_4\mathbf{r}_\downarrow^T\mathbf{r}_\uparrow'\right]\right\}, \tag{E.8}$$

819 which depends on the following scalars,

$$a_1 = \frac{(2\pi)^{N-2}}{\det[\mathbb{B} + \mathbb{B}']}, \tag{E.9}$$

$$c_1 = b_1 - \mathbf{b}_1^T\mathbb{C}\mathbf{b}_1, \qquad\qquad d_1' = 2b_3' - (\mathbf{b}_1')^T\mathbb{C}\mathbf{b}_2' - (\mathbf{b}_2')^T\mathbb{C}\mathbf{b}_1', \tag{E.10}$$

$$c_1' = b_1' - (\mathbf{b}_1')^T\mathbb{C}\mathbf{b}_1', \qquad\qquad f_1 = \mathbf{b}_1^T\mathbb{C}\mathbf{b}_1' + (\mathbf{b}_1')^T\mathbb{C}\mathbf{b}_1, \tag{E.11}$$

$$c_2 = b_2 - \mathbf{b}_2^T\mathbb{C}\mathbf{b}_2, \qquad\qquad f_2 = \mathbf{b}_2^T\mathbb{C}\mathbf{b}_2' + (\mathbf{b}_2')^T\mathbb{C}\mathbf{b}_2, \tag{E.12}$$

$$c_2' = b_2' - (\mathbf{b}_2')^T\mathbb{C}\mathbf{b}_2', \qquad\qquad f_3 = \mathbf{b}_1^T\mathbb{C}\mathbf{b}_2' + (\mathbf{b}_2')^T\mathbb{C}\mathbf{b}_1, \tag{E.13}$$

$$d_1 = 2b_3 - \mathbf{b}_1^T\mathbb{C}\mathbf{b}_2 - \mathbf{b}_2^T\mathbb{C}\mathbf{b}_1, \qquad\qquad f_4 = \mathbf{b}_2^T\mathbb{C}\mathbf{b}_1' + (\mathbf{b}_1')^T\mathbb{C}\mathbf{b}_2, \tag{E.14}$$

820 and on the matrix,

$$\mathbb{C} = (\mathbb{B} + \mathbb{B}')^{-1}. \tag{E.15}$$

821 Next, we Fourier transform Eq. (E.8) according to Eq. (C.9) in order to obtain the ECG ma-
822 trix elements of the two-body density matrix in momentum space:

$$[\tilde{\rho}(\mathbf{p}_1, \mathbf{p}_2)]_{\mathbb{A}\mathbb{A}'} = \frac{1}{(2\pi)^4}\int\cdots\int d^2\mathbf{r}_\uparrow d^2\mathbf{r}_\uparrow' d^2\mathbf{r}_\downarrow d^2\mathbf{r}_\downarrow' [\rho(\mathbf{r}_\uparrow, \mathbf{r}_\uparrow'; \mathbf{r}_\downarrow, \mathbf{r}_\downarrow')]_{\mathbb{A}\mathbb{A}'}\, e^{-i\mathbf{p}_1\cdot(\mathbf{r}_\uparrow' - \mathbf{r}_\uparrow)}e^{-i\mathbf{p}_2\cdot(\mathbf{r}_\downarrow' - \mathbf{r}_\downarrow)}. \tag{E.16}$$

823 By changing variables to $\mathbf{X}_\uparrow = \mathbf{r}_\uparrow' - \mathbf{r}_\uparrow$ and $\mathbf{X}_\downarrow = \mathbf{r}_\downarrow' - \mathbf{r}_\downarrow$, Eq. (E.16) becomes

$$[\tilde{\rho}(\mathbf{p}_1, \mathbf{p}_2)]_{\mathbb{A}\mathbb{A}'} = \frac{a_1}{(2\pi)^4}\int\cdots\int d^2\mathbf{r}_\uparrow d^2\mathbf{r}_\downarrow d^2\mathbf{X}_\uparrow d^2\mathbf{X}_\downarrow \exp[-i(p_1^x X_\uparrow^x + p_1^y X_\uparrow^y + p_2^x X_\downarrow^x + p_2^y X_\downarrow^y)]\times$$

$$\exp\left(\frac{1}{2}\left\{g_1\left[(r_\uparrow^x)^2 + (r_\uparrow^y)^2\right] + g_2\left[(r_\downarrow^x)^2 + (r_\downarrow^y)^2\right] + g_3\left[r_\uparrow^x r_\downarrow^x + r_\uparrow^y r_\downarrow^y\right] + \right.\right.$$

$$\left.\left. h_{\text{temp}}^{(1,x)} r_\uparrow^x + h_{\text{temp}}^{(1,y)} r_\uparrow^y + h_{\text{temp}}^{(2,x)} r_\downarrow^x + h_{\text{temp}}^{(2,y)} r_\downarrow^y + h_{\text{temp}}^{(3)}\right\}\right), \tag{E.17}$$

824 where

$$g_1 = f_1 - c_1 - c_1', \tag{E.18}$$

$$g_2 = f_2 - c_2 - c_2' \,, \tag{E.19}$$

$$g_3 = f_3 + f_4 - d_1 - d_1' \,, \tag{E.20}$$

825 are constant scalars, while

$$h_{\text{temp}}^{(1,i)} = (f_1 - 2c_1')X_\uparrow^i + (f_3 - d_1')X_\downarrow^i \,, \tag{E.21}$$

$$h_{\text{temp}}^{(2,i)} = (f_4 - d_1')X_\uparrow^i + (f_2 - 2c_2')X_\downarrow^i \,, \tag{E.22}$$

$$h_{\text{temp}}^{(3)} = -c_1'[(X_\uparrow^x)^2 + (X_\uparrow^y)^2] - c_2'[(X_\downarrow^x)^2 + (X_\downarrow^y)^2] - d_1'(X_\uparrow^x X_\downarrow^x + X_\uparrow^y X_\downarrow^y) \,, \tag{E.23}$$

826 are temporary functions of the integration variables $\mathbf{X}_\uparrow$ and $\mathbf{X}_\downarrow$ (with $i = x, y$). In similarity to
827 the previous section, the integral over $\mathbf{r}_\downarrow$ can be performed analytically for $g_2 < 0$, and then
828 so can the integral over $\mathbf{r}_\uparrow$ for $4g_1 g_2 - g_3^2 > 0$:

$$\int_{-\infty}^{+\infty} \cdots \int_{-\infty}^{+\infty} dr_\uparrow^x \, dr_\uparrow^y \, dr_\downarrow^x \, dr_\downarrow^y \exp\Bigg( \frac{1}{2} \Big\{ g_1 \big[ (r_\uparrow^x)^2 + (r_\uparrow^y)^2 \big] + g_2 \big[ (r_\downarrow^x)^2 + (r_\downarrow^y)^2 \big] +$$

$$g_3 \big[ r_\uparrow^x r_\downarrow^x + r_\uparrow^y r_\downarrow^y \big] + h_{\text{temp}}^{(1,x)} r_\uparrow^x + h_{\text{temp}}^{(1,y)} r_\uparrow^y + h_{\text{temp}}^{(2,x)} r_\downarrow^x + h_{\text{temp}}^{(2,y)} r_\downarrow^y \Big\} \Bigg)$$

$$= \frac{16\pi^2}{4g_1 g_2 - g_3^2} \exp\Bigg[ -\frac{1/2}{4g_1 g_2 - g_3^2} \times$$

$$\Bigg( g_1 \Big\{ \big[ h_{\text{temp}}^{(2,x)} \big]^2 + \big[ h_{\text{temp}}^{(2,y)} \big]^2 \Big\} + g_2 \Big\{ \big[ h_{\text{temp}}^{(1,x)} \big]^2 + \big[ h_{\text{temp}}^{(1,y)} \big]^2 \Big\} - g_3 \Big\{ h_{\text{temp}}^{(1,x)} h_{\text{temp}}^{(2,x)} + h_{\text{temp}}^{(1,y)} h_{\text{temp}}^{(2,y)} \Big\} \Bigg) \Bigg]$$

$$= \frac{16\pi^2}{t_0} \exp\Bigg( \frac{1}{2t_0} \Big\{ t_1 \big[ (X_\uparrow^x)^2 + (X_\uparrow^y)^2 \big] + t_2 \big[ (X_\downarrow^x)^2 + (X_\downarrow^y)^2 \big] + t_3 \big[ X_\uparrow^x X_\downarrow^x + X_\uparrow^y X_\downarrow^y \big] \Big\} \Bigg) \,, \tag{E.24}$$

829 where we have defined

$$t_0 = 4g_1 g_2 - g_3^2 \,, \tag{E.25}$$

$$t_1 = -(f_4 - d_1')^2 g_1 - (f_1 - 2c_1')^2 g_2 + (f_4 - d_1')(f_1 - 2c_1') g_3 \,, \tag{E.26}$$

$$t_2 = -(f_2 - 2c_2')^2 g_1 - (f_3 - d_1')^2 g_2 + (f_3 - d_1')(f_2 - 2c_2') g_3 \,, \tag{E.27}$$

$$t_3 = -2(f_4 - d_1')(f_2 - 2c_2') g_1 - 2(f_3 - d_1')(f_1 - 2c_1') g_2$$

$$+ [(f_4 - d_1')(f_3 - d_1') + (f_2 - 2c_2')(f_1 - 2c_1')] g_3 \,. \tag{E.28}$$

830 Therefore, Eq. (E.17) can now be written as

$$[\widetilde{\rho}(\mathbf{p}_1, \mathbf{p}_2)]_{\mathbb{AA}'} = \frac{a_1}{(2\pi)^4} \frac{16\pi^2}{t_0} \int \int d^2\mathbf{X}_\uparrow \, d^2\mathbf{X}_\downarrow \exp[-i(p_1^x X_\uparrow^x + p_1^y X_\uparrow^y + p_2^x X_\downarrow^x + p_2^y X_\downarrow^y)] \times$$

$$\exp\Bigg( \frac{1}{2} \Big\{ s_1 \big[ (X_\uparrow^x)^2 + (X_\uparrow^y)^2 \big] + s_2 \big[ (X_\downarrow^x)^2 + (X_\downarrow^y)^2 \big] + s_3 \big[ X_\uparrow^x X_\downarrow^x + X_\uparrow^y X_\downarrow^y \big] \Big\} \Bigg) \,, \tag{E.29}$$

831 which involves

$$s_1 = t_1/t_0 - c_1' \,, \tag{E.30}$$

$$s_2 = t_2/t_0 - c_2' \,, \tag{E.31}$$

$$s_3 = t_3/t_0 - d_1' \,. \tag{E.32}$$

832 At this point, the integral over $\mathbf{X}_\downarrow$ can be carried out analytically for $s_2 < 0$:

$$\int_{-\infty}^{+\infty} \int_{-\infty}^{+\infty} dX_\downarrow^x \, dX_\downarrow^y \exp[-i(p_2^x X_\downarrow^x + p_2^y X_\downarrow^y)] \times$$

$$\exp\left(\frac{1}{2}\left\{s_2\left[(X_\downarrow^x)^2 + (X_\downarrow^y)^2\right] + s_3\left[X_\uparrow^x X_\downarrow^x + X_\uparrow^y X_\downarrow^y\right]\right\}\right)$$
$$= -\frac{2\pi}{s_2}\exp\left(\frac{1}{8s_2}\left\{4\left[(p_2^x)^2 + (p_2^y)^2\right] - s_3^2\left[(X_\uparrow^x)^2 + (X_\uparrow^y)^2\right] + 4is_3(p_2^x X_\uparrow^x + p_2^y X_\uparrow^y)\right\}\right). \quad \text{(E.33)}$$

Subsequently, for $4s_1s_2 - s_3^2 > 0$ we can analytically evaluate the integral over $\mathbf{X}_\uparrow$ as well:

$$\int_{-\infty}^{+\infty}\int_{-\infty}^{+\infty} dX_\uparrow^x\, dX_\uparrow^y \exp[-i(p_1^x X_\uparrow^x + p_1^y X_\uparrow^y)] \times \exp\left\{\frac{1}{2}s_1\left[(X_\uparrow^x)^2 + (X_\uparrow^y)^2\right]\right\} \times$$
$$\exp\left(\frac{1}{8s_2}\left\{4\left[(p_2^x)^2 + (p_2^y)^2\right] - s_3^2\left[(X_\uparrow^x)^2 + (X_\uparrow^y)^2\right] + 4is_3(p_2^x X_\uparrow^x + p_2^y X_\uparrow^y)\right\}\right)$$
$$= -\frac{8\pi s_2}{4s_1s_2 - s_3^2}\exp\left(\frac{2}{4s_1s_2 - s_3^2}\left\{s_2\left[(p_1^x)^2 + (p_1^y)^2\right] + s_1\left[(p_2^x)^2 + (p_2^y)^2\right] - s_3(p_1^x p_2^x + p_1^y p_2^y)\right\}\right).$$
$$\text{(E.34)}$$

Collating and simplifying these results leads to a compact expression for the ECG matrix elements of the momentum-space two-body density matrix for spin-$\uparrow$-spin-$\downarrow$ pairs:

$$[\widetilde{\rho}(\mathbf{p}_1, \mathbf{p}_2)]_{\mathbb{A}\mathbb{A}'} = \frac{a_1}{(2\pi)^4}\frac{16\pi^2}{4g_1g_2 - g_3^2}\left(-\frac{2\pi}{s_2}\right)\left(-\frac{8\pi s_2}{4s_1s_2 - s_3^2}\right) \times$$
$$\exp\left(\frac{2}{4s_1s_2 - s_3^2}\left\{s_2\left[(p_1^x)^2 + (p_1^y)^2\right] + s_1\left[(p_2^x)^2 + (p_2^y)^2\right] - s_3(p_1^x p_2^x + p_1^y p_2^y)\right\}\right)$$
$$= \frac{16a_1}{(4g_1g_2 - g_3^2)(4s_1s_2 - s_3^2)}\exp\left\{\frac{2}{4s_1s_2 - s_3^2}\left[s_2 p_1^2 + s_1 p_2^2 - s_3(p_1^x p_2^x + p_1^y p_2^y)\right]\right\}, \quad \text{(E.35)}$$

with momenta $p_1 \equiv |\mathbf{p}_1|$ and $p_2 \equiv |\mathbf{p}_2|$. We have checked numerically that $g_2$ and $s_2$ are less than zero, while $4g_1g_2 - g_3^2$ and $4s_1s_2 - s_3^2$ are greater than zero, as required. The expectation value $\langle n_\uparrow(\mathbf{p}_1)n_\downarrow(\mathbf{p}_2)\rangle$ can now be evaluated with respect to the ground state in a manner akin to Eq. (D.25). For particles with both opposite spins and opposite momenta ($\mathbf{p}_1 = -\mathbf{p}_2 \equiv \mathbf{p}$) this final result simplifies even further [and notice its similarity to Eq. (D.24)]:

$$[\widetilde{\rho}(\mathbf{p}, -\mathbf{p})]_{\mathbb{A}\mathbb{A}'} = \frac{16a_1}{(4g_1g_2 - g_3^2)(4s_1s_2 - s_3^2)}\exp\left[\frac{2(s_1 + s_2 + s_3)}{4s_1s_2 - s_3^2}p^2\right], \quad \text{(E.36)}$$

with momentum $p \equiv |\mathbf{p}|$. We remark that for clarity, we have used the unsymmetrised basis functions defined by Eq. (D.2) in the above discussion. However, in actuality, these must be antisymmetrised according to the prescription provided at the end of the previous appendix.

# F Bardeen–Cooper–Schrieffer (BCS) Theory

In this appendix, we describe the BCS theoretical treatment for completeness and ease of access. The ensuing derivation of the opposite-momentum pair correlation function, $\mathcal{C}^{(2)}(\mathbf{p}, -\mathbf{p})$ $\equiv \mathcal{C}^{(2)}(p)$, was first performed in Ref. [24] and the results are relevant to Figs. 4 and 6 in the current work.

Within BCS theory, the expectation values in Eqs. (C.2)–(C.4) can be directly evaluated with respect to the ground state by applying the Bogoluibov transformation:

$$c_{\mathbf{p}\uparrow} = u_p \gamma_{\mathbf{p}\uparrow} - v_p \gamma_{-\mathbf{p}\downarrow}^\dagger, \quad \text{(F.1)}$$

$$c_{\mathbf{p}\downarrow} = u_p \gamma_{\mathbf{p}\downarrow} + v_p \gamma_{-\mathbf{p}\uparrow}^\dagger, \quad \text{(F.2)}$$

where

$$u_p^2 = (1 + \varepsilon_p/\xi_p)/2 \,, \tag{F.3}$$

$$v_p^2 = (1 - \varepsilon_p/\xi_p)/2 \,. \tag{F.4}$$

The BCS spectrum of excitations is given by $\xi_p = (\varepsilon_p^2 + \Delta^2)^{1/2}$. Here, $\varepsilon_p = p^2/(2m) - \varepsilon_F$ is the free electron dispersion measured relative to the Fermi energy, and the mean-field value of the superfluid gap is $\Delta = (2\varepsilon_b \varepsilon_F)^{1/2}$ [12]. By replacing the particle creation and annihilation operators ($c_{\mathbf{p}\sigma}^\dagger$, $c_{\mathbf{p}\sigma}$) with fermionic quasiparticle operators ($\gamma_{\mathbf{p}\sigma}^\dagger$, $\gamma_{\mathbf{p}\sigma}$), and then using the fact that the BCS ground state is the quasiparticle vacuum, $\gamma_{\mathbf{p}\sigma}|\Psi_{\mathrm{BCS}}\rangle = 0$, we arrive at

$$\mathcal{C}^{(2)}(p) = \langle c_{\mathbf{p}\uparrow}^\dagger c_{\mathbf{p}\uparrow} c_{-\mathbf{p}\downarrow}^\dagger c_{-\mathbf{p}\downarrow}\rangle - \langle c_{\mathbf{p}\uparrow}^\dagger c_{\mathbf{p}\uparrow}\rangle \langle c_{-\mathbf{p}\downarrow}^\dagger c_{-\mathbf{p}\downarrow}\rangle$$
$$= \mathcal{N}^2 \frac{\Delta^2}{4(\varepsilon_p^2 + \Delta^2)} \,. \tag{F.5}$$

The normalisation factor $\mathcal{N}$ is determined by fixing the single-spin atom number in the non-interacting limit ($\Delta = 0$):

$$N_\uparrow = \int \langle c_{\mathbf{p}\uparrow}^\dagger c_{\mathbf{p}\uparrow}\rangle \, d\mathbf{p} = 2\pi \mathcal{N} \int_0^\infty v_p^2 \, p \, dp \,. \tag{F.6}$$

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
