# Peer review of "When does a Fermi puddle become a Fermi sea? Emergence of Pairing in Two-Dimensional Trapped Mesoscopic Fermi Gases"

_SciPost Physics_

## Round 1 · Referee Report · Zheyu Shi (Referee 1) · 2024-9-15

Report

The manuscript investigates the BCS pairing properties in mesoscopic 2D Fermi gases. The authors employ a stochastic variational approach with an explicitly correlated Gaussian basis to compute eigenenergies and corresponding wave functions for up to six fermions in a harmonic trap. The calculated monopole excitation energy behavior indicates a potential few-body precursor of the many-body normal to superfluid quantum phase transition, as well as the Higgs mode. The resulting pair correlation function reveals that the pairing reaches its maximum below the Fermi surface, which contrasts with experimental measurements for twelve fermions in the trap, suggesting that the Fermi sea emerges in the transition from six to twelve particles.

The emergence of the Fermi sea as particle numbers increase presents a compelling question, closely aligned with recent experiments conducted by S. Jochim's group. The manuscript by Laird et al. offers a valuable contribution to this inquiry from a few-body perspective. The authors present a well-structured paper with a rigorous numerical analysis of the few-body problem, effectively addressing this intriguing phenomenon. I would recommend the manuscript to be published in Scipost Physics, if the authors could address the following comments.

1. On page 6, line 189, the authors wrote “In practical computations, we tune the effective range to large negative values through a shape resonance…”. Could the authors clarify the term "large" in this context? The subsequent calculations utilize a model potential with an effective range of 0.1l2r, which may not be considered large by conventional standards.

2. The critical binding energy (0.953ω for r2d=0.1l2r) is presented towards the end of the manuscript. It would be beneficial to introduce this value earlier in the paper to enhance readers' comprehension of the pair correlation function trends.

3. As a suggestion for further analysis, I think it would be beneficial to examine the behavior of the largest eigenvalue of the two-body density matrix. This metric potentially serves as a better indicator of superfluidity compared to the pairing number defined in the manuscript and Ref.[24]. According to the original work on off-diagonal long-range order (RMP 34, 694 (1962)), this eigenvalue approaches O(N) when superfluidity occurs, with the corresponding eigenfunctions revealing the pairing wavefunction. Given that the authors have already computed the two-body density matrix, it is recommended that they investigate this quantity to enhance the depth of their analysis.

Recommendation

Ask for minor revision

  • validity: -
  • significance: -
  • originality: -
  • clarity: -
  • formatting: -
  • grammar: -

Author:  Emma Laird  on 2024-10-31  [id 4921]

(in reply to Report 1 by Zheyu Shi on 2024-09-15)

We are very grateful for the referee’s careful reading of the manuscript and their insightful feedback. Below, please find our responses to the specific points raised. The numbers of our responses coincide with the numbers in the referee’s report. In our resubmitted manuscript, we have written the associated changes in orange-coloured text to make them immediately obvious.

Point 1:

The referee is correct. According to conventional standards, a "large" effective range (r_2D) would be one which satisfies: h-bar^2/(2m|r_2D|) > h-bar omega_r (noting that our definition of r_2D has units of squared length and is negative in value). This inequality is not satisfied if we substitute in the experimental values of {m, omega_r, r_2D} provided in Ref. [24], and thus, while the effective range in our study is finite and non-negligible, it is not conventionally "large". In consequence, we have altered the concerned sentence to read [page 6, lines 189−190]: "In practical computations, we tune the effective range to non-negligible negative values through a shape resonance...".

Point 2:

We agree with the referee that introducing the value of the critical binding energy earlier (than in Sec. 3.3) would enhance the coherency and comprehension of the text. In order to introduce the concept and value of the critical binding energy in Sec. 3.1, we have altered 5 sentences on pages 9−10 which now read as follows:

Lines 252−253: "At a critical binding energy (denoted by epsilon_b^c) the excitation energy Delta-E reaches a minimum."

Lines 264−267: "In this case, if the ground state has a closed-shell configuration, then the minimum value of Delta-E at the critical binding energy epsilon_b^c will decrease as N increases, eventually reducing to zero in the many-body limit so that pairs are coherently excited without any energy cost [25, 26]."

Lines 267−269: "In this limit, if epsilon_b is increased from zero to epsilon_b^c, then the many-body two-component Fermi gas will become unstable and undergo a second-order phase transition into a superfluid state."

Lines 277−279: "Notably, we find that as the magnitude of the negative effective range increases, the minimum value of Delta-E decreases and shifts to smaller binding energies, i.e., epsilon_b^c decreases."

Lines 284−285: "The value of the critical binding energy for this line is epsilon_b^c = 0.953 h-bar omega_r."

Point 3:

We thank the referee very much for their suggestion. We have conducted an initial analysis in this direction and obtained some preliminary results. In two dimensions, the two-body density matrix in Eq. (6), rho(r_1, r’_1; r_2, r’_2), is an eight-dimensional array. Therefore, we need to reduce the number of degrees of freedom before diagonalising it to obtain its eigenvalues (occupation numbers). In the first instance, we have done this by following Ref. [39] which treated a very similar problem in three dimensions. The idea is to first transform from the co-ordinates of the individual particles to the centre-of-mass (R, R’) and relative (r, r’) co-ordinates of the two spin-up-spin-down pairs, and to then set the relative distance vectors equal to each other, r = r’. Setting r = r’ is somewhat artificial; however, one then obtains a reduced two-body density matrix, rho_red(R, R’), which has the same number of degrees of freedom as the one-body density matrix, and thus, is diagonalisable once it has been decomposed into partial-wave contributions. Due to setting r = r’, the largest eigenvalue of rho_red(R, R’) does not directly indicate the onset of superfluidity. Nevertheless, the "condensate fraction" may be estimated by comparing the magnitudes of the eigenvalues of rho_red(R, R’) — e.g., see Eq. (16) in Ref. [39]. By analysing the reduced two-body density matrix in two dimensions, we have obtained plots of the occupation numbers and "condensate fraction" as a function of the two-body binding energy, akin to Fig. 11 in Ref. [39]. However, since rho_red(R, R’) only measures non-local correlations between spin-up-spin-down pairs that are characterised by the same relative distance vector, we believe this approach should be more carefully examined and compared with alternative methods. Hence, while the referee has given us a great idea, we have reached the conclusion that it is beyond the scope of the current work. We believe it deserves a more thorough and detailed investigation which should form the focus of a future paper. In the current paper, on the other hand, we are instead more focused on observables that are measurable in experiments.

Anonymous on 2024-10-31  [id 4923]

(in reply to Emma Laird on 2024-10-31 [id 4921])

The authors have adequately addressed my comments. I now recognize that diagonalizing the two-body density matrix is a more complex task than I initially presumed. I appreciate the authors' clarification on this matter.

---

## Round 1 · Referee Report · Anonymous (Referee 2) · 2024-10-12

Report

The manuscripts presents a detailed theoretical study of few fermions in a 2D harmonic trap, interacting via an attractive short-range two-body potential, and the emergence of pairing when the number of particles is increased. The study aims to explain the experimental observations of Ref.[24] , where a system of 12 fermions already shows correlations typical of a Fermi sea. The present study is limited to 3+3 fermions, but it provides clear hints of the emergence of correlations as the Fermi sphere is gradually formed. The authors employ almost exact numerical techniques to calculate ground state and excited state energies and correlation functions in momentum space aiming to characterize pairing effects in such mesoscopic systems.
The results are interesting and in my opinion deserve publication in SciPost as they can have an important impact on the ongoing experimental research on the precise manipulation of quantum systems composed by few interacting fermions. Furthermore, the paper is well structured and clearly written, with the main results clearly presented and discussed while all technical details are left to a large body of appendices. In my opinion the manuscript should be published once the authors have considered the points I'm listing below.
-) Figure 2 contains results of the lowest excited state as a function of the strength of the attractive interaction. In the case of the 3+3 system, the non monotonous behavior of the excitation energy is explained as the result of pairing correlations. Being however the excitation energy the result of an energy difference between the excited and the ground state, it is not clear how the ground and excited state energies would behave separately. It would be maybe useful to add a figure showing the energy of the ground state for the 1+1, 2+2 and 3+3 system as a function of the binding energy epsilon_b.
-)In figure 6 the fraction of paired fermions is shown as a function of the binding energy. In the text it is stated that the maximum number of pairs is 3 (for the 3+3 system) . In the figure caption, however, it is not clear whether the maximum of N_pair should be 3 or if what is plotted is the fraction of pairs which saturates at 1. One would also expect that for large enough attraction each fermion pair would bind forming a tightly bound dimer similar to the BEC regime in macroscopic systems. Is this something that emerges from the calculations? Could the authors maybe comment on this point?
-)In section 3.3 lines 417-419 the authors write "In the many-body limit N\to\infty, the system remains in the normal state for epsilon_b<<hbar omega_r and undergoes a quantum phase transition to a superfluid state...". In my opinion this statement is incorrect: in the N\to\infty limit the Fermi energy is much larger than the level separation hbar omega_r and the same occurs to the gap. As a consequence at T=0 the system is always in the superfluid state. The authors should clarify what they mean by many-body limit. A similar line of reasoning is probably present also at the beginning of the fourth paragraph in section 3.1 (starting at line 261), where it is stated that the minimum of the excitation energy Delta E becomes soft in the thermodynamic limit. Should not this be the finite gap of BCS theory? A clarification is also required here.

Recommendation

Ask for minor revision

  • validity: -
  • significance: -
  • originality: -
  • clarity: -
  • formatting: -
  • grammar: -

Author:  Emma Laird  on 2024-10-31  [id 4922]

(in reply to Report 2 on 2024-10-12)

We thank the referee for their useful and insightful feedback, which we believe has helped us to improve our manuscript. Below, please find our responses to the specific points raised. We number the referee’s comments from top to bottom as 1, 2, 3. In our resubmitted manuscript, we have written the associated changes in blue-coloured text to make them immediately obvious.

Point 1 (top):

We agree with the referee that it would be helpful to show the ground- and excited-state energies used to calculate the excitation energy separately. In Fig. 2 on page 8, we have created a new middle panel (b) where the ground- and excited-state energies from panel (a) [i.e., for 1+1, 2+2, and 3+3 fermions at zero effective range] are plotted separately as a function of the two-body binding energy. This makes it clear how the different dependencies of the excitation energies on the interaction strength arise in panel (a). To accommodate this change, we have reworked the caption of Fig. 2 on page 8 and edited/added two sentences in the first paragraph of Sec. 3.1 "Excitation Spectrum" on page 7 [lines 218–221 and 224–226].

Point 2 (middle):

We thank the referee for pointing out our lack of clarity regarding the "number of pairs" which saturates at 3 (for 3+3 fermions) in the strong-interaction limit, versus the "paired fraction" which saturates at 1; we always mean the number of pairs. To correct this, we have renamed Sec. 3.3 from "Paired Fraction" to "Number of Pairs" and replaced the phrase "paired fraction" in the Abstract with "number of pairs". We have also added a new sentence to the caption of Fig. 6 to remove any remaining ambiguity: "The maximum possible number of pairs is N_pair = 3.". (In Sec. 3.3 the text itself is clear on the fact that N_pair is the number of pairs with a maximal value of 3.)

The referee is correct to expect that for much stronger interactions than those shown, epsilon_b > 2 h-bar omega_r, all the fermions would form tightly bound bosonic dimers, reminiscent of the deep BEC regime in macroscopic systems. We have added a new sentence to this effect in Sec. 3.3 [lines 413–416]. (We had also mentioned this earlier on page 12 [footnote 8] where we first discuss our considered range of binding energies.) The reason why Fig. 6 does not show any red data points for epsilon_b > 2 h-bar omega_r is that for such strong binding energies, it is challenging to properly model the tight composite bosonic wave functions, and thus, to obtain fully numerically converged energies and structural properties within a reasonable time frame. We now explain this point in a new footnote [11] in Sec. 3.3.

Point 3 (bottom):

Indeed, our mention of the many-body limit was vague and hence confusing. We are very grateful to the referee for bringing this important point to our attention. In free space where BCS theory applies, the many-body limit is typically approached by increasing both the number of particles, N -> infinity, and the system volume, V -> infinity, in such a way that the density n = N/V remains constant. For our scenario where omega_r is fixed, we instead have n -> infinity when N -> infinity. In this case, the many-body two-component Fermi gas remains in the normal state for epsilon_b << h-bar omega_r and undergoes a quantum phase transition to a superfluid state at epsilon_b^c. If we wish to make our system amenable to BCS theory, then we could keep n constant by reducing the trap strength omega_r while increasing N. In that case, the trapping frequency would vanish (omega_r -> 0) for N -> infinity, which means the condition epsilon_b << h-bar omega_r would never be satisfied in the many-body limit. Consequently, a superfluid state with a finite gap would always exist at zero temperature for any non-vanishing interaction strength. We have clarified this point about the many-body limit in both sections of the text highlighted by the referee: in Sec. 3.1 "Excitation Spectrum" on page 9 [lines 263–269 and footnote 7] and in Sec. 3.3 "Number of Pairs" [lines 423–428].

---

## Round 2 · Author Response

Dear Editor,

Please find our resubmitted manuscript attached to this correspondence in PDF format. We have responded to each referee in detail below their respective reports. Edits to the manuscript are written in different coloured text so that they are immediately obvious: orange text is associated with our response to Referee Report #1, blue text is associated with our response to Referee Report #2, and any additional corrections not associated with either report are written in green text.

Thank you for your time and kind regards,
Emma (Laird)

---

## Round 2 · List of Changes

1. Middle initials have been included in the authors' names, where appropriate [page 1].
2. Suburbs have been included in the affiliation addresses [page 1].
3. In Sec. 2, the word "large" which appears in reference to the effective range has been changed to "non-negligible" [line 189].
4. To clarify the concept of the critical binding energy, we have edited/added five sentences throughout Sec. 3.1 [lines 252-253, 264-267, 267-269, 277-279, 284-285].
5. A new middle panel (b) has been added to Fig. 2 which shows the ground- and first-excited-state energies separately, and the figure caption has been reworked accordingly [page 8].
6. Two sentences have been edited/added in the first paragraph of Sec. 3.1 to reflect the change in Fig. 2 [lines 218-221 and 224-226].
7. To distinguish between the "paired fraction" and the "number of pairs", the phrase "paired fraction" in the Abstract has been replaced with "number of pairs" [page 1], Sec. 3.3 has been renamed from "Paired Fraction" to "Number of Pairs" [pages 1 and 15], and one sentence has been added to the caption of Fig. 6 [page 17].
8. To provide more information on our computational limitations in the BEC regime, footnote 8 has been edited [page 12], footnote 11 has been added [page 16], and one sentence has been added to Sec. 3.3 [lines 413-416].
9. To clarify what we mean by the "many-body limit", one sentence [lines 263-264] has been rewritten and a new footnote [7] has been added to Sec. 3.1, and three sentences [lines 423-428] have been rewritten in Sec. 3.3.
10. Four names have been added to the Acknowledgements section [lines 531-533].
11. An error has been corrected in Eq. (A.10) of Appendix A.
12. Some figures have been repositioned within the text after the above edits were made.

---

## Editorial Decision

accepted_in_target_journal